# XI-LEARNING: SUCCESSOR FEATURE TRANSFER LEARNING FOR GENERAL REWARD FUNCTIONS

## ABSTRACT

Transfer in Reinforcement Learning aims to improve learning performance on target tasks using knowledge from experienced source tasks. Successor features (SF) are a prominent transfer mechanism in domains where the reward function changes between tasks. They reevaluate the expected return of previously learned policies in a new target task and to transfer their knowledge. A limiting factor of the SF framework is its assumption that rewards linearly decompose into successor features and a reward weight vector. We propose a novel SF mechanism, $\xi$-learning, based on learning the cumulative discounted probability of successor features. Crucially, $\xi$-learning allows to reevaluate the expected return of policies for general reward functions. We introduce two $\xi$-learning variations, prove its convergence, and provide a guarantee on its transfer performance. Experimental evaluations based on $\xi$-learning with function approximation demonstrate the prominent advantage of $\xi$-learning over available mechanisms not only for general reward functions, but also in the case of linearly decomposable reward functions.

## 1 INTRODUCTION

Reinforcement Learning (RL) successfully addressed many complex problems such as playing computer games, chess, and even Go with superhuman performance (Mnih et al., 2015; Silver et al., 2018). These impressive results are possible thanks to a vast amount of interactions of the RL agent with its environment/task. Such strategy is unsuitable in settings where the agent has to perform and learn at the same time. Consider, for example, a care giver robot in a hospital that has to learn a new task, such as a new route to deliver meals. In such a setting, the agent can not collect a vast amount of training samples but has to adapt quickly instead. Transfer learning aims to provide mechanisms quickly to adapt agents in such settings (Taylor and Stone, 2009; Lazaric, 2012; Zhu et al., 2020). The rationale is to use knowledge from previously encountered *source* tasks for a new *target* task to improve the learning performance on the target task. The previous knowledge can help reducing the amount of interactions required to learn the new optimal behavior. For example, the care giver robot could reuse knowledge about the layout of the hospital it learned in previous source tasks (e.g. guiding a person) to learn to deliver meals.

The Successor Feature (SF) and General Policy Improvement (GPI) framework (Barreto et al., 2020) is a prominent transfer learning mechanism for tasks where only the reward function differs. Its basic premise is that the rewards which the RL agent tries to maximize are defined based on a low-dimensional feature descriptor $\phi \in \mathbb{R}^n$. For our care-giver robot this could be ID's of beds or rooms that it is visiting, in difference to its high-dimensional visual state input from a camera. The rewards are then computed not based on its visual input but on the ID's of the beds or rooms that it visits. The expected cumulative discounted successor features ($\psi$) are learned for each behavior that the robot learned in the past. It represents the dynamics in the feature space that the agent experiences for a behavior. This corresponds to the rooms or beds the care-giver agent would visit if using the behavior. This representation of feature dynamics is independent from the reward function. A behavior learned in a previous task and described by this SF representation can be directly re-evaluated for a different reward function. In a new task, i.e. for a new reward function, the GPI procedure re-evaluates the behaviors learned in previous tasks for it. It then selects at each state the behavior of a previous task if it improves the expected reward. This allows to reuse behaviors learned in previous source tasks

---

Source code at `https://tinyurl.com/3xuzxff3`

for a new target task. A similar transfer strategy can also be observed in the behavior of humans (Momennejad et al., 2017; Momennejad, 2020; Tomov et al., 2021) .

The classical SF&GPI framework (Barreto et al., 2017; 2018) makes the assumption that rewards $r$ are a linear composition of the features $\phi \in \mathbb{R}^n$ via a reward weight vector $\mathbf{w}_i \in \mathbb{R}^n$ that depends on the task $i$: $r_i = \phi^\top \mathbf{w}_i$. This assumption allows to effectively separate the feature dynamics of a behavior from the rewards and thus to re-evaluate previous behaviors given a new reward function, i.e. a new weight vector $\mathbf{w}_j$. Nonetheless, this assumption also restricts successful application of SF&GPI only to problems where such a linear decomposition is possible. We investigate the application of the SF&GPI framework to general reward functions: $r_i = R_i(\phi)$ over the feature space. We propose to learn the cumulative discounted probability over the successor features, named $\xi$-function, and refer to the proposed framework as $\xi$-learning. Our work is related to Janner et al. (2020); Touati and Ollivier (2021), and brings two important additional contributions. First, we provide mathematical proof of the convergence of $\xi$-learning. Second, we demonstrate how $\xi$-learning can be used for meta-RL, using the $\xi$-function to re-evaluate behaviors learned in previous tasks for a new reward function $R_j$. Furthermore, $\xi$-learning can also be used to transfer knowledge to new tasks using GPI.

The contribution of our paper is three-fold:

- We introduce a new RL algorithm, $\xi$-learning, based on a cumulative discounted probability of successor features, and two variants of its update operator.
- We provide theoretical proofs of the convergence of $\xi$-learning to the optimal policy and for a guarantee of its transfer learning performance under the GPI procedure.
- We experimentally compare $\xi$-learning in tasks with linear and general reward functions, and for tasks with discrete and continuous features to standard Q-learning and the classical SF framework, demonstrating the interest and advantage of $\xi$-learning.

## 2 BACKGROUND

### 2.1 REINFORCEMENT LEARNING

RL investigates algorithms to solve multi-step decision problems, aiming to maximize the sum over future rewards (Sutton and Barto, 2018). RL problems are modeled as *Markov Decision Processes* (MDPs) which are defined as a tuple $M \equiv (\mathcal{S}, \mathcal{A}, p, R, \gamma)$, where $\mathcal{S}$ and $\mathcal{A}$ are the state and action set. An agent transitions from a state $s_t$ to another state $s_{t+1}$ using action $a_t$ at time point $t$ collecting a reward $r_t$: $s_t \xrightarrow{a_t, r_t} s_{t+1}$. This process is stochastic and the transition probability $p(s_{t+1}|s_t, a_t)$ describes which state $s_{t+1}$ is reached. The reward function $R$ defines the scalar reward $r_t = R(s_t, a_t, s_{t+1}) \in \mathbb{R}$ for the transition. The goal in an MDP is to maximize the expected return $G_t = \mathrm{E}\left[\sum_{k=0}^\infty \gamma^k R_{t+k}\right]$, where $R_t = R(S_t, A_t, S_{t+1})$. The discount factor $\gamma \in [0, 1)$ weights collected rewards by discounting future rewards stronger. RL provides algorithms to learn a policy $\pi : \mathcal{S} \to \mathcal{A}$ defining which action to take in which state to maximise $G_t$.

Value-based RL methods use the concept of value functions to learn the optimal policy. The state-action value function, called Q-function, is defined as the expected future return taking action $a_t$ in $s_t$ and then following policy $\pi$:

$$Q^\pi(s_t, a_t) = \mathrm{E}_\pi \left\{ r_t + \gamma r_{t+1} + \gamma^2 r_{t+2} + \ldots \right\} = \mathrm{E}_\pi \left\{ r_t + \gamma \max_{a_{t+1}} Q^\pi(S_{t+1}, a_{t+1}) \right\} . \quad (1)$$

The Q-function can be recursively defined following the Bellman equation such that the current Q-value $Q^\pi(s_t, a_t)$ depends on the maximum Q-value of the next state $Q^\pi(s_{t+1}, a_{t+1})$. The optimal policy for an MDP can then be expressed based on the Q-function, by taking at every step the maximum action: $\pi^*(s) \in \mathrm{argmax}_a Q^*(s, a)$.

The optimal Q-function can be learned using a temporal difference method such as Q-learning (Watkins and Dayan, 1992). Given a transition $(s_t, a_t, r_t, s_{t+1})$, the Q-value is updated according to:

$$Q_{k+1}(s_t, a_t) = Q_k(s_t, a_t) + \alpha_k \left( r_t + \max_{a_{t+1}} Q_k(s_{t+1}, a_{t+1}) - Q_k(s_t, a_t) \right) , \quad (2)$$

where $\alpha_k \in (0, 1]$ is the learning rate at iteration $k$.

## 2.2 Transfer Learning and the SF&GPI Framework

We are interested in the transfer learning setting where the agent has to solve a set of tasks $\mathcal{M} = \{M_1, M_2, \ldots, M_m\}$, that in our case differ only in their reward function. The *Successor Feature* (SF) framework provides a principled way to perform transfer learning (Barreto et al., 2017; 2018). SF assumes that the reward function can be decomposed into a linear combination of features $\phi \in \Phi \subset \mathbb{R}^n$ and a reward weight vector $\mathbf{w}_i \in \mathbb{R}^n$ that is defined for a task $M_i$:

$$r_i(s_t, a_t, s_{t+1}) \equiv \phi(s_t, a_t, s_{t+1})^\top \mathbf{w}_i . \tag{3}$$

We refer to such reward functions as *linear reward functions*. Since the various tasks differ only in their reward functions, the features are the same for all tasks in $\mathcal{M}$.

Given the decomposition above, it is also possible to rewrite the Q-function into an expected discounted sum over future features $\psi^{\pi_i}(s, a)$ and the reward weight vector $\mathbf{w}_i$:

$$
\begin{aligned}
Q_i^{\pi_i}(s, a) &= \mathrm{E}\left\{ r_t + \gamma^1 r_{t+1} + \gamma^2 r_{t+2} + \ldots \right\} = \mathrm{E}\left\{ \phi_t^\top \mathbf{w}_i + \gamma^1 \phi_{t+1}^\top \mathbf{w}_i + \gamma^2 \phi_{t+2}^\top \mathbf{w}_i + \ldots \right\} \\
&= \mathrm{E}\left\{ \sum_{k=0}^\infty \gamma^k \phi_{t+k} \right\}^\top \mathbf{w}_i \equiv \psi^{\pi_i}(s, a)^\top \mathbf{w}_i .
\end{aligned}
\tag{4}
$$

This decouples the dynamics of the policy $\pi_i$ in the feature space of the MDP from the expected rewards for such features. Thus, it is now possible to evaluate the policy $\pi_i$ in a different task $M_j$ using a simple multiplication of the weight vector $\mathbf{w}_j$ with the $\psi$-function: $Q_j^{\pi_i}(s, a) = \psi^{\pi_i}(s, a)^\top \mathbf{w}_j$. Interestingly, the $\psi$ function also follows the Bellman equation:

$$\psi^\pi(s, a) = \mathrm{E}\left\{ \phi_{t+1} + \gamma \psi^\pi(s_{t+1}, \pi(s_{t+1})) | s_t, a_t \right\} , \tag{5}$$

and can therefore be learned with conventional RL methods. Moreover, (Lehnert and Littman, 2019) showed the equivalence of SF-learning to Q-learning.

Being in a new task $M_j$ the *Generalized Policy Improvement* (GPI) can be used to select the action over all policies learned so far that behaves best:

$$\pi(s) \in \operatorname*{argmax}_a \max_i Q_j^{\pi_i}(s, a) = \operatorname*{argmax}_a \max_i \psi^{\pi_i}(s, a)^\top \mathbf{w}_j . \tag{6}$$

(Barreto et al., 2018) proved that under the appropriate conditions for optimal policy approximates, the policy constructed in (6) is close to the optimal one, and their difference is upper-bounded:

$$\|Q^* - Q^\pi\|_\infty \leq \frac{2}{1 - \gamma}\left( \|r - r_i\|_\infty + \min_j \|r_i - r_j\|_\infty + \epsilon \right) , \tag{7}$$

where $\|f - g\|_\infty = \max_{s,a} |f(s, a) - g(s, a)|$. For an arbitrary reward function $r$ the result can be interpreted in the following manner. Given the arbitrary task $M$, we identify the theoretically closest possible linear reward task $M_i$ with $r_i$. For this theoretically closest task, we search the linear task $M_j$ in our set of task $\mathcal{M}$ (from which we also construct the GPI optimal policy (6)) which is closest to it. The upper bound between $Q^*$ and $Q$ is then defined by 1) the difference between task $M$ and the theoretically closest possible linear task $M_i$: $\|r - r_i\|_\infty$; and by 2) the difference between theoretical task $M_i$ and the closest task $M_j$: $\min_j \|r_i - r_j\|_\infty$. If our new task $M$ is also linear then $r = r_i$ and the first term in (7) would vanish.

Very importantly, this result shows that the SF framework will only provide a good approximation of the true Q-function if the reward function in a task can be represented using a linear decomposition. If this is not the case then the error in the approximation increases with the distance between the true reward function $r$ and the best linear approximation of it $r_i$ as stated by $\|r - r_i\|_\infty$.

## 3 Method: $\xi$-learning

### 3.1 Definition and Foundations of $\xi$-learning

The goal of this paper is to investigate the application of SF&GPI to tasks with *general reward functions* $R : \Phi \mapsto \mathbb{R}$ over state features $\phi \in \Phi$:

$$r(s_t, a_t, s_{t+1}) \equiv R(\phi(s_t, a_t, s_{t+1})) = R(\phi_t) , \tag{8}$$

where we define $\phi_t \equiv \phi(s_t, a_t, s_{t+1})$. Under this assumption the Q-function can not be linearly decomposed into a part that describes feature dynamics and one that describes the rewards as in the linear SF framework (4). To overcome this issue, we propose to define the expected cumulative discounted probability of successor features or $\xi$-function, which is going to be the central mathematical object of the paper, as:

$$\xi^\pi(s, a, \phi) = \sum_{k=0}^\infty \gamma^k p(\phi_{t+k} = \phi | s_t = s, a_t = a; \pi), \tag{9}$$

where $p(\phi_{t+k} = \phi | s_t = s, a_t = a; \pi)$, or in short $p(\phi_{t+k} = \phi | s_t, a_t; \pi)$, is the probability density function of the features at time $t + k$, following policy $\pi$ and conditioned to $s$ and $a$ being the state and action at time $t$ respectively. Note that $\xi^\pi$ depends not only on the policy $\pi$ but also on the state transition (constant through the paper). With the definition of the $\xi$-function, the Q-function rewrites (this is compatible with SFQL in the linear reward case, see Appendix A.6):

$$Q^\pi(s_t, a_t) = \sum_{k=0}^\infty \gamma^k \mathbb{E}_{p(\phi_{t+k}|s_t, a_t; \pi)} \{R(\phi_{t+k})\} = \sum_{k=0}^\infty \gamma^k \int_\Phi p(\phi_{t+k} = \phi | s_t, a_t; \pi) R(\phi) \mathrm{d}\phi$$
$$= \int_\Phi R(\phi) \sum_{k=0}^\infty \gamma^k p(\phi_{t+k} = \phi | s_t, a_t; \pi) \mathrm{d}\phi = \int_\Phi R(\phi) \xi^\pi(s_t, a_t, \phi) \mathrm{d}\phi. \tag{10}$$

Depending on the reward function $R$, there are several $\xi$-functions that correspond to the same $Q$ function. Formally, this is an equivalence relationship, and the quotient space has a one-to-one correspondence with the $Q$-function space.

**Proposition 1.** *(Equivalence between functions $\xi$ and Q) Let $\mathcal{Q} = \{Q : \mathcal{S} \times \mathcal{A} \to \mathbb{R} \ s.t. \ \|Q\|_\infty < \infty\}$. Let $\sim$ be defined as $\xi_1 \sim \xi_2 \Leftrightarrow \int_\Phi R\xi_1 = \int_\Phi R\xi_2$. Then, $\sim$ is an equivalence relationship, and there is a bijective correspondence between the quotient space $\Xi_\sim$ and $\mathcal{Q}$.*

**Corollary 1.** *The bijection between $\Xi_\sim$ and $\mathcal{Q}$ allows to induce a norm $\| \cdot \|_\sim$ into $\Xi_\sim$ from the supremum norm in $\mathcal{Q}$, with which $\Xi_\sim$ is a Banach space (since $\mathcal{Q}$ is Banach with $\| \cdot \|_\infty$):*

$$\|\xi\|_\sim = \sup_{s,a} \left| \int_\Phi R(\phi)\xi(s, a, \phi) d\phi \right| = \sup_{s,a} |Q(s, a)| = \|Q\|_\infty. \tag{11}$$

Similar to the Bellman equation for the Q-function, we can define a Bellman operator for the $\xi$-function, denoted by $T_\xi$, as:

$$T_\xi(\xi^\pi) = p(\phi_t = \phi | s_t, a_t) + \gamma \mathbb{E}_{p(s_{t+1}, a_{t+1} | s_t, a_t; \pi)} \{\xi^\pi(s_{t+1}, a_{t+1}, \phi)\}. \tag{12}$$

As in the case of the $Q$-function, we can use $T_\xi$ to construct a contractive operator:

**Proposition 2.** *($\xi$-learning has a fixed point) The operator $T_\xi$ is well-defined w.r.t. the equivalence $\sim$, and therefore induces an operator $T_\sim$ defined over $\Xi_\sim$. $T_\sim$ is contractive w.r.t. $\| \cdot \|_\sim$. Since $\Xi_\sim$ is Banach, $T_\sim$ has a unique fixed point and iterating $T_\sim$ starting anywhere converges to that point.*

In other words, successive applications of the operator $T_\sim$ converge towards the class of optimal $\xi$ functions $[\xi^*]$ or equivalently to an optimal $\xi$ function defined up to an additive function $k$ satisfying $\int_\Phi k(s, a, \phi)R(\phi)\mathrm{d}\phi = 0, \forall(s, a) \in \mathcal{S} \times \mathcal{A}$ (i.e. $k \in \text{Ker}(\xi \to \int_\Phi R\xi)$).

While these two results state (see Appendix A for the proofs) the theoretical links to standard Q-learning formulations, the $T_\xi$ operator defined in (12) is not usable in practice, because of the expectation. In the next section, we define the optimisation iterate, prove its convergence, and provide two variants to perform the $\xi$ updates.

### 3.2 $\xi$-LEARNING ALGORITHMS

In order to learn the $\xi$-function, we introduce the $\xi$-*learning* update operator, which is an off-policy temporal difference method analogous to Q-learning. Given a transition $(s_t, a_t, s_{t+1}, \phi_t)$ the $\xi$-learning update operator is defined as:

$$\xi_{k+1}^\pi(s_t, a_t, \phi) \leftarrow \xi_k^\pi(s_t, a_t, \phi) + \alpha_k [p(\phi_t = \phi | s_t, a_t) + \gamma \xi_k^\pi(s_{t+1}, \bar{a}_{t+1}, \phi) - \xi_k^\pi(s_t, a_t, \phi)], \tag{13}$$

where $\bar{a}_{t+1} = \text{argmax}_a \int_\Phi R(\phi)\xi^\pi(s_{t+1}, a, \phi)\mathrm{d}\phi$.

The following is one of the main results of the manuscript, stating the convergence of $\xi$-learning:

**Theorem 1.** *(Convergence of $\xi$-learning) For a sequence of state-action-feature $\{s_t, a_t, s_{t+1}, \phi_t\}_{t=0}^{\infty}$ consider the $\xi$-learning update given in (13). If the sequence of state-action-feature triples visits each state, action infinitely often, and if the learning rate $\alpha_k$ is an adapted sequence satisfying the Robbins-Monro conditions:*

$$\sum_{k=1}^{\infty} \alpha_k = \infty, \qquad \sum_{k=1}^{\infty} \alpha_k^2 < \infty \tag{14}$$

*then the sequence of function classes corresponding to the iterates converges to the optimum, which corresponds to the optimal Q-function to which standard Q-learning updates would converge to:*

$$[\xi_n] \to [\xi^*] \quad with \quad Q^*(s, a) = \int_{\Phi} R(\phi)\xi^*(s, a, \phi)d\phi. \tag{15}$$

The proof is provided in Appendix A and follows the same flow as for Q-learning.

The previous theorem provides convergence guarantees under the assumption that either $p(\phi_t = \phi | s_t, a_t; \pi)$ is known, or an unbiased estimate can be constructed. We propose two different ways to approximate $p(\phi_t = \phi | s_t, a_t; \pi)$ from a given transition $(s_t, a_t, s_{t+1}, \phi_t)$ so as to perform the $\xi$-update (13). The first instance is a model-free version and detailed in the following section. A second instance uses a one-step SF model, called *One-Step Model-based (MB) $\xi$-learning*, which is further described in Sec. B.

**Model-free (MF) $\xi$-Learning:** MF $\xi$-learning uses the same principle as standard model-free temporal difference learning methods. The update assumes for a given transition $(s_t, a_t, s_{t+1}, \phi_t)$ that the probability for the observed feature is $p(\phi = \phi_t | s_t, a_t) = 1$. Whereas for all other features $(\forall \phi' \in \Phi, \phi' \neq \phi_t)$ the probability is $p(\phi' = \phi_t | s_t, a_t) = 0$, see Appendix D for continuous features. The resulting updates are:

$$\begin{aligned} \phi = \phi_t: \quad & \xi^{\pi}(s_t, a_t, \phi) & \leftarrow & \quad (1 - \alpha)\xi^{\pi}(s_t, a_t, \phi) + \alpha \left(1 + \gamma \xi^{\pi}(s_{t+1}, \bar{a}_{t+1}, \phi)\right) \\ \phi' \neq \phi_t: \quad & \xi^{\pi}(s_t, a_t, \phi') & \leftarrow & \quad (1 - \alpha)\xi^{\pi}(s_t, a_t, \phi') + \alpha \gamma \xi^{\pi}(s_{t+1}, \bar{a}_{t+1}, \phi'). \end{aligned} \tag{16}$$

Due to the stochastic update of the $\xi$-function and if the learning rate $\alpha \in (0, 1]$ discounts over time, the $\xi$-update will learn the true probability of $p(\phi = \phi_t | s_t, a_t)$. A potential problem with the MF procedure is that it might induce a high variance when the true feature probabilities are not binary.

### 3.3 META $\xi$-LEARNING

After discussing $\xi$-learning on a single task and showing its theoretical convergence, we can now investigate how it can be applied in transfer learning. Similar to the linear SF framework the $\xi$-function allows to reevaluate a policy learned for task $M_i$, $\xi^{\pi_i}$, in a new environment $M_j$:

$$Q_j^{\pi_i}(s, a) = \int_{\Phi} R_j(\phi)\xi^{\pi_i}(s, a, \phi)d\phi. \tag{17}$$

This allows us to apply GPI in (6) for arbitrary reward functions in a similar manner to what was proposed for linear reward functions in (Barreto et al., 2018). We extend the GPI result to the $\xi$-learning framework as follows:

**Theorem 2.** *(Generalised policy improvement in $\xi$-learning) Let $\mathcal{M}$ be the set of tasks, each one associated to a (possibly different) weighting function $R_i \in L^1(\Phi)$. Let $\xi^{\pi_i^*}$ be a representative of the optimal class of $\xi$-functions for task $M_i$, $i \in \{1, \ldots, I\}$, and let $\tilde{\xi}^{\pi_i}$ be an approximation to the optimal $\xi$-function, $\|\xi^{\pi_i^*} - \tilde{\xi}^{\pi_i}\|_{R_i} \leq \varepsilon, \forall i$. Then, for another task $M$ with weighting function $R$, the policy defined as:*

$$\pi(s) = \arg\max_a \max_i \int_{\Phi} R(\phi)\tilde{\xi}^{\pi_i}(s, a, \phi)d\phi, \tag{18}$$

*satisfies:*

$$\|\xi^* - \xi^{\pi}\|_R \leq \frac{2}{1 - \gamma}(\min_i \|R - R_i\|_{p(\phi|s,a)} + \varepsilon), \tag{19}$$

*where $\|f\|_g = \sup_{s,a} \int_{\Phi} |f \cdot g| \, d\phi$.*

The proof is provided in Appendix A.

## 4 EXPERIMENTS

We evaluated $\xi$-learning in two environments. The first has discrete features. It is a modified version of the object collection task by Barreto et al. (2017) having more complex features to allow general reward functions. See Appendix E.1 for experimental results in the original environment. The second environment, the racer environment, evaluates the agents in tasks with continuous features.

### 4.1 DISCRETE FEATURES - OBJECT COLLECTION ENVIRONMENT

**Environment:** The environment consist of 4 rooms (Fig. 1 - a). The agent starts an episode in position S and has to learn to reach the goal position G. During an episode, the agent can collect objects to gain further rewards. Each object has 2 properties: 1) color: orange or blue, and 2) form: box or triangle. The state space is a high-dimensional vector $s \in \mathbb{R}^{112}$. It encodes the agent's position using a $10 \times 10$ grid of two-dimensional Gaussian radial basis functions. Moreover, it includes a memory about which object has been already collected. Agents can move in 4 directions. The features $\phi \in \Phi = \{0,1\}^5$ are binary vectors. The first 2 dimensions encode if an orange or a blue object was picked up. The 2 following dimensions encode the form. The last dimension encodes if the agent reached goal G. For example, $\phi^\top = [1,0,1,0,0]$ encodes that the agent picked up an orange box.

**Tasks:** Each agent learns sequentially 300 tasks which differ in their reward for collecting objects. We compared agents in two settings: either in tasks with linear or general reward functions. For each linear task $\mathcal{M}_i$, the rewards $r = \phi^\top \mathbf{w}_i$ are defined by a linear combination of features and a weight vector $\mathbf{w}_i \in \mathbb{R}^5$. The weights $w_{i,k}$ for the first 4 dimensions define the rewards for collecting an object with a specific property. They are randomly sampled from a uniform distribution: $w_{i,k} \sim \mathcal{U}(-1,1)$. The final weight defines the reward for reaching the goal position which is $w_{i,5} = 1$ for each task. The general reward functions are sampled by assigning a different reward to each possible combination of object properties $\phi_j \in \Phi$ using uniform sampling: $R_i(\phi_j) \sim \mathcal{U}(-1,1)$, such that picking up an orange box might result in a reward of $R_i(\phi^\top = [1,0,1,0,0]) = 0.23$.

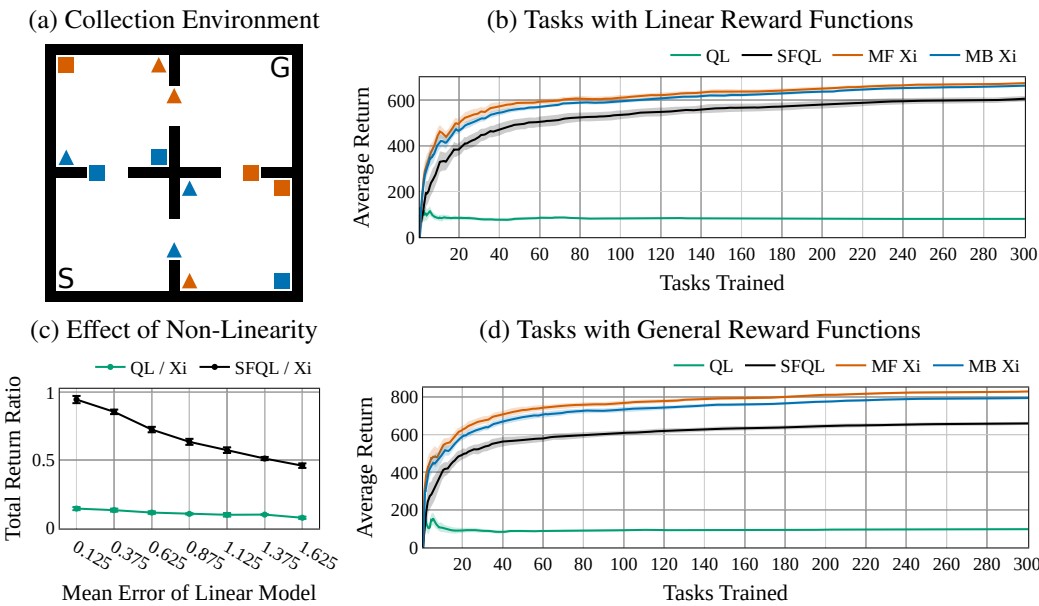

Figure 1: In the (a) object collection environment, $\xi$-learning reached the highest average reward per task for (b) linear, and (d) general reward functions. The average over 10 runs per algorithm and the standard error of the mean are depicted. (c) The performance difference between $\xi$-learning and SFQL is stronger for general reward tasks that have high non-linearity, i.e. where a linear reward model yields a high error. SFQL can only reach less than $50\%$ of MF $\xi$-learning's performance in tasks with a mean linear reward model error of 1.625.

**Agents:** We compared $\xi$-learning to Q-learning (QL), and classical SF Q-learning (SFQL) (Barreto et al., 2017). All agents use function approximation for their state-action functions (Q, $\psi$, or $\xi$-function). An independent linear mapping is used to map the values from the state for each of the 4 actions. As the features are discrete, the $\xi$-function and $\hat{p}_\phi$-model are approximated by an independent mapping for each action and possible feature $\phi \in \Phi$. The Q-value $Q(s, a)$ for the $\xi$-agents (Eq. 10) is computed by: $Q^\pi(s, a) = \sum_{\phi \in \Phi} R(\phi) \xi^\pi(s, a, \phi)$. The reward functions of each task are given to the $\xi$-agents. For SFQL, the sampled reward weights $\mathbf{w}_i$ were given in tasks with linear reward functions. For general reward functions, a linear model $r = \phi^\top \tilde{\mathbf{w}}_i$ approximating the rewards was learned for each task and its weights $\tilde{\mathbf{w}}_i$ given to SFQL (see Appendix C.3 for details). Each tasks was executed for $20,000$ steps, and the average performance over 10 runs per algorithm was measured. We performed a grid-search over the parameters of each agent, reporting here the performance of the parameters with the highest total reward over all tasks.

**Results:** $\xi$-learning outperformed SFQL and QL for tasks with linear and general reward functions (Fig. 1 - b; d). MF showed a slight advantage over MB $\xi$-learning in both settings. We further studied the effect non-linearity of general reward functions on the performance of classical SF compared to $\xi$-learning by evaluating them in tasks with different levels of non-linearity. We sampled general reward functions that resulted in different levels of mean absolute model error if they are linearly approximated with $\min_{\tilde{\mathbf{w}}} |r(\phi) - \phi^\top \tilde{\mathbf{w}}|$. We trained SFQL and MF $\xi$-learning in each of these conditions on 300 tasks and measured the ratio between the total return of SFQL and MF $\xi$ (Fig. 1). The relative performance of SFQL compared to MF $\xi$ reduces with higher non-linearity of the reward functions. For reward functions that are nearly linear (mean error of $0.125$), both have a similar performance. Whereas, for reward functions that are difficult to model with a linear relation (mean error of $1.625$) SFQL reaches only less than $50\%$ of the performance of $\xi$-learning. This follows SFQL's theoretical limitation in (7) and shows the advantage of $\xi$ learning over SFQL in non-linear reward tasks.

## 4.2 Continuous Features - Racer Environment

**Environment and Tasks:** We further evaluated the agents in an environment with continuous features (Fig. 2 - a). The agent is randomly placed in the environment and has to drive around for 200 timesteps before the episode ends. Similar to a car, the agent has an orientation and momentum, so that it can only drive straight, or in a right or left curve. The agent reappears on the opposite side if it exits one side. The distance to 3 markers are provided as features $\phi \in \mathbb{R}^3$. Rewards depend on the distances $r = \sum_{k=1}^{3} r_k \phi_k$, where each component $r_k$ has 1 or 2 preferred distances defined by Gaussian functions. For each of the 37 tasks, the number of Gaussians and their properties $(\mu, \sigma)$ are randomly sampled for each feature dimension. Fig. 2 (a) shows a reward function with dark areas depicting higher rewards. The agent has to learn to drive around in such a way as to maximize its trajectory over positions with high rewards. The state space is a high-dimensional vector $s \in \mathbb{R}^{120}$ encoding the agent's position and orientation. As before, the 2D position is encoded using a $10 \times 10$ grid of two-dimensional Gaussian radial basis functions. Similarly, the orientation is also encoded using 20 Gaussian radial basis functions.

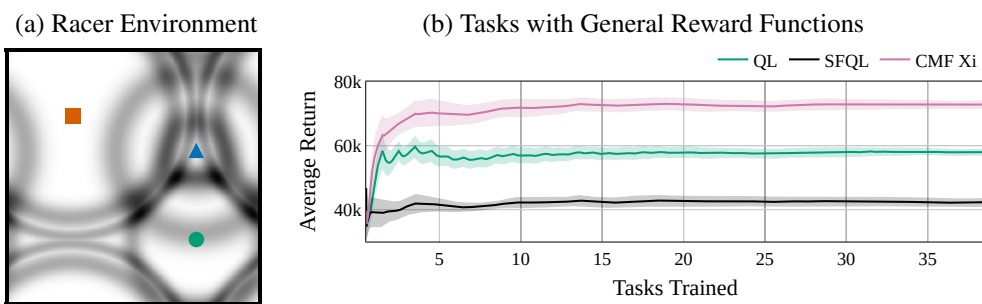

Figure 2: (a) Example of a reward function for the racer environment based on distances to its 3 markers. (b) $\xi$-learning reaches the highest average reward per task. SFQL yields a performance even below QL as it is not able to model the reward function with its linear combination of weights and features. The average over 10 runs per agent and the standard error of the mean are depicted.

**Agents:** We introduce a MF $\xi$-agent for continuous features (CMF $\xi$) (Appendix D.2.3). CMF $\xi$ discretizes each feature dimension $\phi_k \in [0, 1]$ in 11 bins with the bin centers: $X = \{0.0, 0.1, \ldots, 1.0\}$. It learns for each dimension $k$ and bin $i$ the $\xi$-value $\xi_k^\pi(s, a, X_i)$. Q-values (Eq. 10) are computed by: $Q^\pi(s, a) = \sum_{k=1}^{3} \sum_{i=1}^{11} r_k(X_i) \xi_k^\pi(s, a, X_i)$. SFQL learns $\psi$ for the continuous, non-discretized feature space. It received an approximated weight vector $\tilde{\mathbf{w}}_i$ that was trained before the task started on several uniformly sampled features and rewards.

**Results:** $\xi$-learning reached the highest performance of all agents (Fig. 2 - b) outperforming QL and SFQL. SFQL reaches only a low performance below QL, because it is not able to sufficiently well approximate the general reward functions with its linear reward model. This shows the advantage of $\xi$-learning over SFQL in environments with general reward functions.

## 5 DISCUSSION

**Performance of $\xi$-learning compared to classical SF&GPI:** $\xi$-learning allows to disentangle the dynamics of policies in the feature space of a task from the associated reward, see (10). The experimental evaluation in tasks with general reward functions (Fig. 1 - d, and Fig. 2) shows that $\xi$-learning can therefore successfully apply GPI to transfer knowledge from learned tasks to new ones. Given a general reward function it can re-evaluate successfully learned policies for knowledge transfer. Instead, classical SFQL based on a linear decomposition (3) can not be directly applied given a general reward function. In this case a linear approximation has to be learned which shows inferior performance to $\xi$-learning that directly uses the true reward function.

$\xi$-learning also shows an increased performance over SFQL in environments with linear reward functions (Fig. 1 - a). This effect can not be attributed to differences in their computation of a policy's expected return as both are correct (Appendix A.6). A possible explanation is that $\xi$-learning reduces the complexity for the function approximation of the $\xi$-function compared to the $\psi$-function in SFQL.

**Continuous Feature Spaces:** For tasks with continuous features (racer environment), $\xi$-learning used successfully a discretization of each feature dimension, and learned the $\xi$-values independently for each dimension. This strategy is viable for reward functions that are cumulative over the feature dimensions: $r(\phi) = \sum_k r_k \phi_k$. The Q-value can be computed by summing over the independent dimensions and the bins $X$: $Q^\pi(s, a) = \sum_k \sum_{x \in X} r_k(x) \xi^\pi(s, a, x)$. For more general reward functions, the space of all feature combinations would need to be discretized, which grows exponentially with each new dimension. As a solution the $\xi$-function could be directly defined over the continuous feature space, but this yields some problems. First, the computation of the expected return requires an integral $Q(s, a) = \int_{\phi \in \Phi} R(\phi) \xi(s, a, \phi)$ over features instead of a sum, which is a priori intractable. Second, the representation and training of the $\xi$-function, which would be defined over a continuum thus increasing the difficulty of approximating the function. Janner et al. (2020) and Touati and Ollivier (2021) propose methods that might allow to represent a continuous $\xi$-function, but it is unclear if they converge and if they can be used for transfer learning.

**Learning of Features:** In principle, classical SF&GPI can also optimize general reward functions if features and reward weights are learned. This is possible if the learned features describe the non-linear effects in the reward functions. Nonetheless, learning of features adds further challenges and shows to reduce performance. Barreto et al. (2017) learns features from observations sampled from several tasks before the SF&GPI starts. Therefore, novel non-linearities potentially introduced at later tasks are not well represent by the learned features. If instead features are learned alongside the SF&GPI procedure, the problem on how to coordinate both learning processes needs to be investigated. Importantly, $\psi$-functions for older tasks would become unusable for the GPI procedure on newer task, because the feature representation changed between them.

Moreover, our replication (Sec. E.1) of the object collection task from Barreto et al. (2017) shows the performance of learned features is below the performance of given features. MF Xi reaches a final average reward per task of $850$ with given features and reward functions. The best performance of SFQL with learned features only reaches a final performance of $575$ (Fig. 2 in (Barreto et al., 2017)).

In summary, if features and reward functions are known then $\xi$-learning outperforms SFQL. And using given features and reward functions is natural for many applications as these are often known, for example in robotic tasks where they are usually manually designed (Akalin and Loutfi, 2021).

**Computational Complexity:** The improved performance of SFQL and $\xi$-learning over QL in the transfer learning setting comes at the cost of an increased computational complexity. The GPI procedure (6) of both approaches requires to evaluate at each step the $\psi^{\pi_i}$-function or $\xi^{\pi_i}$-function over all previous experienced tasks in $\mathcal{M}$. As a consequence, the computational complexity increases linearly with each new environment that is added. A solution is to apply GPI only over a subset of learned policies. Nonetheless, an open question is still how to optimally select this subset.

# 6 RELATED WORK

**Transfer Learning:** Transfer methods in RL can be generally categorized according to the type of tasks between which transfer is possible and the type of transferred knowledge (Taylor and Stone, 2009; Lazaric, 2012; Zhu et al., 2020). In the case of SF&GPI which $\xi$-learning is part of, tasks only differ in their reward functions. The type of knowledge that is transferred are policies learned in source tasks which are re-evaluated in the target task and recombined using the GPI procedure. A natural use-case for $\xi$-learning are continual problems (Khetarpal et al., 2020) where an agent has continually adapt to changing tasks, which are in our setting different reward functions.

**Successor Features:** SF are based on the concept of *successor representations* (Dayan, 1993; Momennejad, 2020). Successor representations predict the future occurrence of all states for a policy in the same manner as SF for features. Their application is restricted to low-dimensional state spaces using tabular representations. SF extended them to domains with high-dimensional state spaces (Kulkarni et al., 2016; Zhang et al., 2017; Barreto et al., 2017; 2018), by predicting the future occurrence of low-dimensional features that are relevant to define the return. Several extensions to the SF framework have been proposed. One direction aims to learn appropriate features from data such as by optimally reconstruct rewards (Barreto et al., 2017), using the concept of mutual information (Hansen et al., 2019), or the grouping of temporal similar states (Madjiheurem and Toni, 2019). Another direction is the generalization of the $\psi$-function over policies (Borsa et al., 2018) analogous to universal value function approximation (Schaul et al., 2015). Similar approaches use successor maps (Madarasz, 2019), goal-conditioned policies (Ma et al., 2020), or successor feature sets (Brantley et al., 2021). Other directions include their application to POMDPs (Vértes and Sahani, 2019), combination with max-entropy principles (Vertes, 2020), or hierarchical RL (Barreto et al., 2021). In difference to $\xi$-learning all these approaches build on the assumption of linear reward functions, whereas $\xi$-learning allows the SF&GPI framework to be used with general reward functions. Nonetheless, most of the extensions for linear SF can be combined with $\xi$-learning.

**Model-based RL:** SF represent the dynamics of a policy in the feature space that is decoupled from the rewards allowing to reevaluate them under different reward functions. It shares therefore similar properties with model-based RL (Lehnert and Littman, 2019). In general, model-based RL methods learn a one-step model of the environment dynamics $p(s_{t+1}|s_t, a_t)$. Given a policy and an arbitrary reward function, rollouts can be performed using the learned model to evaluate the return. In practice, the rollouts have a high variance for long-term predictions rendering them ineffective. Recently, (Janner et al., 2020) proposed the $\gamma$-model framework that learns to represent $\xi$-values in continuous domains. Nonetheless, the application to transfer learning is not discussed and no convergence is proven as for $\xi$-learning. This is the same case for the forward-backward MPD representation proposed in Touati and Ollivier (2021). (Tang et al., 2021) also proposes to decouple the dynamics in the state space from the rewards, but learn an internal representation of the rewards. This does not allow to reevaluate an policy to a new reward function without relearning the mapping.

# 7 CONCLUSION

The introduced $\xi$-learning framework learns the expected cumulative discounted probability of successor features which disentangles the dynamics of a policy in the feature space of a task from the expected rewards. This allows $\xi$-learning to reevaluate the expected return of learned policies for general reward functions and to use it for transfer learning utilizing GPI. We proved that $\xi$-learning converges to the optimal policy, and showed experimentally its improved performance over Q-learning and the classical SF framework for tasks with linear and general reward functions.

## ETHICS STATEMENT

$\xi$-learning and its associated optimization algorithms represent general RL procedures similar to Q-learning. Their potential negative societal impact depends on their application domains which range over all possible societal areas in a similar manner as for other general RL procedures.

Beyond the topic of the paper, we did our best to cite the relevant literature and to fairly compare with previous ideas, concepts and methods. To that aim, all agents are trained and evaluated within the same software environment, and under the very same experimental settings.

## REPRODUCIBILITY STATEMENT

In order to ensure high changes of reproducibility we provided lots of details of the method and experiments associated to the paper. In particular, we have provided the proofs for all mathematical results announced in the main paper (see Appendix A). These constitute the theoretical foundation of the proposed $\xi$-learning methodology. Secondly, we have provided all experimental details (methods, and environments) required for reproducing our experiments, namely: appendix C for the object collection and D for the racer environment respectively. In addition, we provide additional results in appendix E, to completely illustrate the interest of the proposed method. Finally, we provided an anonymous link to the source code, so that reviewers can run it if necessary.

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

## A  THEORETICAL PROOFS

### A.1  PROOF OF PROPOSITION 1

Let us start by recalling the original statement in the main paper.

**Proposition 1.** *(Equivalence between functions $\xi$ and Q) Let $\mathcal{Q} = \{Q : \mathcal{S} \times \mathcal{A} \to \mathbb{R} \ s.t. \ \|Q\|_\infty < \infty\}$. Let $\sim$ be defined as $\xi_1 \sim \xi_2 \Leftrightarrow \int_\Phi R\xi_1 = \int_\Phi R\xi_2$. Then, $\sim$ is an equivalence relationship, and there is a bijective correspondence between the quotient space $\Xi_\sim$ and $\mathcal{Q}$.*

*Proof.* We will proof the statements sequentially.

$\sim$ **is an equivalence relationship:**  To prove this we need to demonstrate that $\sim$ is symmetric, reciprocal and transitive. The three are quite straightforward since: $\xi \sim \xi, \forall \xi, \xi \sim \eta \Leftrightarrow \eta \sim \xi, \forall \xi, \eta$ and $\xi \sim \eta, \eta \sim \nu \Rightarrow \xi \sim \nu$.

**Bijective correspondence:**  To prove the bijectivity, we will first prove that it is injective, then surjective. Regarding the injectivity: $[\xi] \neq [\eta] \Rightarrow Q_\xi \neq Q_\eta$, we prove it by contrapositive:

$$Q_\xi = Q_\eta \Rightarrow \int_\Phi R(\phi)\xi(s,a,\phi)\mathrm{d}\phi = \int_\Phi R(\phi)\eta(s,a,\phi)\mathrm{d}\phi \Rightarrow [\xi] = [\eta]. \tag{20}$$

In order to prove the surjectivity, we start from a function $Q \in \mathcal{Q}$ and select an arbitrary $\xi \in \Xi$, then the following function:

$$\xi_Q(s,a,\phi) = \frac{Q(s,a)}{\int_\Phi R(\bar{\phi})\xi(s,a,\bar{\phi})\mathrm{d}\bar{\phi}}\xi(s,a,\phi) \tag{21}$$

satisfies that $\xi_Q \in \Xi$ and that $\int_\Phi R(\phi)\xi_Q(s,a,\phi)\mathrm{d}\phi = Q(s,a), \forall(s,a) \in \mathcal{S} \times \mathcal{A}$. We conclude that there is a bijective correspondence between the elements of $\Xi_\sim$ and of $\mathcal{Q}$.  $\square$

### A.2  PROOF OF COROLLARY 1

Let us recall the result:

**Corollary 1.** *The bijection between $\Xi_\sim$ and $\mathcal{Q}$ allows to induce a norm $\|\cdot\|_\sim$ into $\Xi_\sim$ from the supremum norm in $\mathcal{Q}$, with which $\Xi_\sim$ is a Banach space (since $\mathcal{Q}$ is Banach with $\|\cdot\|_\infty$):*

$$\|\xi\|_\sim = \sup_{s,a}\left|\int_\Phi R(\phi)\xi(s,a,\phi)d\phi\right| = \sup_{s,a}|Q(s,a)| = \|Q\|_\infty \ , \tag{22}$$

*Proof.* The norm induced in the quotient space is defined from the correspondence between $\Xi_\sim$ and $\mathcal{Q}$ and is naturally defined as in the previous equation. The norm is well defined since it does not depend on the class representative. Therefore, all the metric properties are transferred, and $\Xi_\sim$ is immediately Banach with the norm $\|\cdot\|_\sim$.  $\square$

### A.3  PROOF OF PROPOSITION 2

Let's restate the result:

**Proposition 2.** *($\xi$-learning has a fixed point) The operator $T_\xi$ is well-defined w.r.t. the equivalence $\sim$, and therefore induces an operator $T_\sim$ defined over $\Xi_\sim$. $T_\sim$ is contractive w.r.t. $\|\cdot\|_\sim$. Since $\Xi_\sim$ is Banach, $T_\sim$ has a unique fixed point and iterating $T_\sim$ starting anywhere converges to that point.*

*Proof.* We prove the statements above one by one:

**The operator $T_\sim$ is well defined:** Let us first recall the definition of the operator $T_\xi$ in (12), where we removed the dependency on $\pi$ for simplicity:

$$T_\xi(\xi) = p(\phi_t = \phi|s_t, a_t) + \gamma E_{p(s_{t+1}, a_{t+1}|s_t, a_t)} \{\xi(s_{t+1}, a_{t+1}, \phi)\}$$

Let $\xi_1, \xi_2 \in [\xi]$ two different representatives of class $[\xi]$, we can write:

$$
\begin{aligned}
\int_\Phi R(\phi)&(T_\xi(\xi_1)(s, a, \phi) - T_\xi(\xi_2)(s, a, \phi)) d\phi \\
&= \int_\Phi R(\phi) \int_{\mathcal{S}} p(s', a'|s, a)\gamma(\xi_1(s', a', \phi) - \xi_2(s', a', \phi)) ds' d\phi \\
&= \gamma \int_{\mathcal{S}} p(s', a'|s, a) \int_\Phi R(\phi)(\xi_1(s', a', \phi) - \xi_2(s', a', \phi)) d\phi ds' \\
&= 0
\end{aligned}
\tag{23}
$$

because $\xi_1, \xi_2 \in [\xi]$. Therefore the operator $T_\sim([\xi]) = T_\xi(\xi)$ is well defined in the quotient space, since the image of class does not depend on the function chosen to represent the class.

**Contractive operator $T_\sim$:** The contractiveness of $T_\sim$ can be proven directly:

$$
\begin{aligned}
\|T_\sim([\xi]) - T_\sim([\eta])\|_\sim &= \sup_{s,a} \left| \int_\Phi R(\phi) \left( p(\phi|s, a) + \gamma E_{p(s', a'|s, a)}\{\xi(s', a', \phi)\} \right. \right. \\
&\qquad\qquad \left. \left. -p(\phi|s, a) - \gamma E_{p(s', a'|s, a)}\{\eta(s', a', \phi)\} \right) d\phi \right| \\
&= \gamma \sup_{s,a} \left| \int_\Phi R(\phi) E_{p(s', a'|s, a)}\{\xi(s', a', \phi) - \eta(s', a', \phi)\} d\phi \right| \\
&= \gamma \sup_{s,a} E_{p(s', a'|s, a)} \left\{ \left| \int_\Phi R(\phi)(\xi(s', a', \phi) - \eta(s', a', \phi)) d\phi \right| \right\} \\
&\leq \gamma \sup_{s', a'} \left| \int_\Phi R(\phi)(\xi(s', a', \phi) - \eta(s', a', \phi)) d\phi \right| \\
&= \gamma \|[\xi] - [\eta]\|_\sim
\end{aligned}
\tag{24}
$$

The contractiveness of $T_\sim$ can also be understood as being inherited from the standard Bellmann operator on $Q$. Indeed, given a $\xi$ function, one can easily see that applying the standard Bellman operator to the $Q$ function corresponding to $\xi$ leads to the $Q$ function corresponding to $T_\sim([\xi])$.

**Fixed point of $T_\sim$:** To conclude the proof, we use the fact that any contractive operator on a Banach space, in our case: $T_\sim : \Xi_\sim \to \Xi_\sim$ has a unique fixed point $[\xi^*]$, and that for any starting point $[\xi_0]$, the sequence $[\xi_n] = T_\sim([\xi_{n-1}])$ converges to $[\xi^*]$ w.r.t. to the corresponding norm $\|[\xi]\|_\sim$. □

## A.4 PROOF OF THEOREM 1

These two propositions will be useful to prove that the $\xi$ learning iterates converge in $\Xi_\sim$. Let us restate the definition of the operator from (13):

$$\xi_{k+1}^\pi(s_t, a_t, \phi) \leftarrow \xi_k^\pi(s_t, a_t, \phi) + \alpha_k [p(\phi_t = \phi|s_t, a_t; \pi) + \gamma \xi_k^\pi(s_{t+1}, \bar{a}_{t+1}, \phi) - \xi_k^\pi(s_t, a_t, \phi)]$$

and the theoretical result:

**Theorem 1.** *(Convergence of $\xi$-learning) For a sequence of state-action-feature $\{s_t, a_t, s_{t+1}, \phi_t\}_{t=0}^\infty$ consider the $\xi$-learning update given in (13). If the sequence of state-action-feature triples visits each state, action infinitely often, and if the learning rate $\alpha_k$ is an adapted sequence satisfying the Robbins-Monro conditions:*

$$\sum_{k=1}^\infty \alpha_k = \infty, \qquad \sum_{k=1}^\infty \alpha_k^2 < \infty \tag{25}$$

*then the sequence of function classes corresponding to the iterates converges to the optimum, which corresponds to the optimal Q-function to which standard Q-learning updates would converge to:*

$$[\xi_n] \to [\xi^*] \quad with \quad Q^*(s,a) = \int_\Phi R(\phi)\xi^*(s,a,x)d\phi. \tag{26}$$

*Proof.* The proof re-uses the flow of the proof used for Q-learning (Tsitsiklis, 1994). Indeed, we rewrite the operator above as:

$$\xi^\pi_{k+1}(s_t,a_t,\phi) \leftarrow \xi^\pi_k(s_t,a_t,\phi) + \alpha_k \left[T_\xi(\xi^\pi_k)(s_t,a_t,\phi) - \xi^\pi_k(s_t,a_t,\phi) + \varepsilon(s_t,a_t,\phi)\right]$$

with $\varepsilon$ defined as:

$$\varepsilon(s_t,a_t,\phi) = \begin{array}{l} p(\phi_t = \phi|s_t,a_t;\pi) + \gamma\xi^\pi_k(s_{t+1},\bar{a}_{t+1},\phi) \\ -\mathrm{E}\left\{p(\phi_t = \phi|s_t,a_t;\pi) + \gamma\xi^\pi_k(s_{t+1},\bar{a}_{t+1},\phi)\right\}. \end{array}$$

Obviously $\varepsilon$ satisfies $\mathrm{E}\{\varepsilon\} = 0$, which, together with the contractiveness of $T_\sim$, is sufficient to demonstrate the convergence of the iterative procedure as done for Q-learning. In our case, the optimal function $\xi^*$ is defined up to an additive kernel function $\kappa \in \mathrm{Ker}$. The correspondence with the optimal Q learning function is a direct application of the correspondence between the $\xi$- and Q-learning problems. $\qquad\square$

### A.5 PROOF OF THEOREM 2

Let us restate the result.

**Theorem 2.** *(Generalised policy improvement in $\xi$-learning) Let $\mathcal{M}$ be the set of tasks, each one associated to a (possibly different) weighting function $R_i \in L^1(\Phi)$. Let $\xi^{\pi^*_i}$ be a representative of the optimal class of $\xi$-functions for task $M_i$, $i \in \{1,\ldots,I\}$, and let $\tilde{\xi}^{\pi_i}$ be an approximation to the optimal $\xi$-function, $\|\xi^{\pi^*_i} - \tilde{\xi}^{\pi_i}\|_{R_i} \le \varepsilon, \forall i$. Then, for another task $M$ with weighting function $R$, the policy defined as:*

$$\pi(s) = \arg\max_a \max_i \int_\Phi R(\phi)\tilde{\xi}^{\pi_i}(s,a,\phi)d\phi, \tag{27}$$

*satisfies:*

$$\|\xi^* - \xi^\pi\|_R \le \frac{2}{1-\gamma}(\min_i \|R - R_i\|_{p(\phi|s,a)} + \varepsilon), \tag{28}$$

*where $\|f\|_g = \sup_{s,a} \int_\Phi |f \cdot g| \, d\phi$.*

*Proof.* The proof is stated in two steps. First, we exploit the proof of Proposition 1 of (Barreto et al., 2017), and in particular (13) that states:

$$\|Q^* - Q^\pi\|_\infty \le \frac{2}{1-\gamma}\left(\sup_{s,a}|r(s,a) - r_i(s,a)| + \varepsilon\right), \quad \forall i \in \{1,\ldots,I\}, \tag{29}$$

where $Q^*$ and $Q^\pi$ are the Q-functions associated to the optimal and $\pi$ policies in the environment $R$. The conditions on the Q functions required in the original proposition are satisfied because the $\xi$-functions satisfy them, and there is an isometry between Q and $\xi$ functions.

Because the above inequality is true for all training environments $i$, we can rewrite as:

$$\|Q^* - Q^\pi\|_\infty \le \frac{2}{1-\gamma}\left(\min_i \sup_{s,a}|r(s,a) - r_i(s,a)| + \varepsilon\right). \tag{30}$$

We now realise that, in the case of $\xi$-learning, the reward functions rewrite as:

$$r(s,a) = \int_\Phi R(\phi)p(\phi|s,a)\mathrm{d}\phi \qquad r_i(s,a) = \int_\Phi R_i(\phi)p(\phi|s,a)\mathrm{d}\phi, \tag{31}$$

and therefore we have:

$$\begin{aligned} \sup_{s,a}|r(s,a) - r_i(s,a)| &= \sup_{s,a}\left|\int_\Phi (R(\phi) - R_i(\phi))p(\phi|s,a)\mathrm{d}\phi\right| \\ &\le \sup_{s,a}\int_\Phi |R(\phi) - R_i(\phi)|\, p(\phi|s,a)\mathrm{d}\phi \\ &= \|R - R_i\|_{p(\phi|s,a)} \end{aligned} \tag{32}$$

Similarly, due to the isometry between $\xi$ and Q-learning, i.e. Proposition 2, we can write that:

$$\|\xi^* - \xi^\pi\|_R = \|[\xi^*] - [\xi^\pi]\|_\sim = \|Q^* - Q^\pi\|_\infty$$

$$\leq \frac{2}{1-\gamma} \left( \min_i \sup_{s,a} |r(s,a) - r_i(s,a)| + \varepsilon \right) \tag{33}$$

$$\leq \frac{2}{1-\gamma} (\min_i \|R - R_i\|_{p(\phi|s,a)} + \varepsilon),$$

which proves the generalised policy improvement for $\xi$-learning. $\qquad\square$

### A.6 RELATION BETWEEN CLASSICAL SF AND $\xi$-LEARNING FOR LINEAR REWARD FUNCTIONS

In the case of linear reward functions, i.e. where assumption (3) holds, it is possible to show that $\xi$-learning can be reduced to classical SF. Classical SF represents therefore a specific case of $\xi$-learning under this assumption.

**Theorem 3.** *(Equality of classical SF and $\xi$-learning for linear reward functions) Given the assumption that reward functions are linearily decomposable with*

$$r_i(s_t, a_t, s_{t+1}) \equiv \phi(s_t, a_t, s_{t+1})^\top \mathbf{w}_i ,$$

*where $\phi \in \mathbb{R}^n$ are features for a transition and $\mathbf{w}_i \in \mathbb{R}^n$ are the reward weight vector of task $m_i \in \mathcal{M}$, then the classical SF and $\xi$-learning framework are equivalent.*

*Proof.* We start with the definition of the Q-value according to $\xi$-learning from (10). After replacing the reward function $R_i$ with our linear assumption, the definition of the Q-function according to classical SF with the $\psi$-function can be recovered:

$$Q_i(s, a) = \int_\Phi \xi^\pi(s_t, a_t, \phi) R_i(\phi) \mathrm{d}\phi$$

$$= \int_\Phi \xi^\pi(s_t, a_t, \phi) \phi^\top \mathbf{w}_i \mathrm{d}\phi$$

$$= \mathbf{w}_i^\top \int_\Phi \sum_{k=0}^\infty \gamma^k p(\phi_{t+k} = \phi | s_t = s, a_t = a; \pi) \phi \, \mathrm{d}\phi$$

$$= \mathbf{w}_i^\top \sum_{k=0}^\infty \gamma^k \int_\Phi p(\phi_{t+k} = \phi | s_t = s, a_t = a; \pi) \phi \, \mathrm{d}\phi$$

$$= \mathbf{w}_i^\top \sum_{k=0}^\infty \gamma^k \mathrm{E} \{\phi_{t+k}\}$$

$$= \mathrm{E} \left\{ \sum_{k=0}^\infty \gamma^k \phi_{t+k} \right\}^\top \mathbf{w}_i = \psi(s, a)^\top \mathbf{w}_i .$$

$\qquad\square$

Please note, although both methods are equal in terms of their computed values, how these are represented and learned differs between them. Thus, it is possible to see a performance difference of the methods in the experimental results where $\xi$-learning outperforms SFQL in our environments.

## B ONE-STEP SF MODEL-BASED (MB) $\xi$-LEARNING

Besides the MF $\xi$-learning update operator (16), we introduce a second $\xi$-learning procedure called *One-step SF Model-based (MB) $\xi$-Learning* that attempts to reduce the variance of the update. To do so, MB $\xi$-learning estimates the distribution over the successor features over time. Let $\tilde{p}(\phi_t = \phi | s_t, a_t; \pi)$ denote the current estimate of the feature distribution. Given a transition $(s_t, a_t, s_{t+1}, \phi_t)$ the model is updated according to:

$$\begin{aligned} \phi = \phi_t : \quad & \tilde{p}_\phi(\phi | s_t, a_t; \pi) \quad \leftarrow \quad \tilde{p}_\phi(\phi | s_t, a_t; \pi) + \beta \left(1 - \tilde{p}_\phi(\phi | s_t, a_t; \pi)\right) \\ \phi' \neq \phi_t : \quad & \tilde{p}_\phi(\phi' | s_t, a_t; \pi) \quad \leftarrow \quad \tilde{p}_\phi(\phi' | s_t, a_t; \pi) - \beta \tilde{p}_\phi(\phi' | s_t, a_t; \pi) , \end{aligned}$$

where $\beta \in [0, 1]$ is the learning rate. After updating the model $\tilde{p}_\phi$, it can be used for the $\xi$-update as defined in (13). Since the learned model $\tilde{p}_\phi$ is independent from the reward function and from the policy, it can be learned and used over all tasks.

## C  EXPERIMENTAL DETAILS: OBJECT COLLECTION ENVIRONMENT

The object collection environment (Fig. 1 - a) was briefly introduced in Section 4.1. This section provides a formal description.

### C.1  ENVIRONMENT

The environment is a continuous two-dimensional area in which the agent moves. The position of the agent is a point in the 2D space: $(x, y) \in [0, 1]^2$. The action space of the agent consists of four movement directions: $A = \{$up, down, left, right$\}$. Each action changes the position of the agent in a certain direction and is stochastic by adding a Gaussian noise. For example, the action for going right updates the position according to $x_{t+1} = x_t + \mathcal{N}(\mu = 0.05, \sigma = 0.005)$. If the new position ends in a wall (black areas in Fig. 1 - a) that have a width of $0.04$) or outside the environment, the agent is set back to its current position. Each environment has 12 objects. Each object has two properties with two possible values: color (orange, blue) and shape (box, triangle). If the agent reaches an object, it collects the object which then disappears. The objects occupy a circular area with radius $0.04$. At the beginning of an episode the agent starts at location S with $(x, y)_{\mathrm{S}} = (0.05, 0.05)$. An episode ends if the agent reaches the goal area G which is at position $(x, y)_{\mathrm{G}} = (0.86, 0.86)$ and has a circular shape with radius $0.1$. After an episode the agent is reset to the start position S and all collected objects reappear.

The state space of the agents consist of their position in the environment and the information about which objects they already collected during an episode. Following (Barreto et al., 2017), the position is encoded using a radial basis function approach. This upscales the agent's $(x, y)$ position to a high-dimensional vector $s_{\mathrm{pos}} \in \mathbb{R}^{100}$ providing a better signal for the function approximation of the different functions such as the $\psi$ or $\xi$-function. The vector $s_{\mathrm{pos}}$ is composed of the activation of two-dimensional Gaussian functions based on the agents position $(x, y)$:

$$s_{\mathrm{pos}} = \exp\left(-\frac{(x - c_{j,1})^2 + (y - c_{j,2})^2}{\sigma}\right), \tag{34}$$

where $c_j \in \mathbb{R}^2$ is the center of the $j^{\mathrm{th}}$ Gaussian. The centers are laid out on a regular $10 \times 10$ grid over the area of the environment. The state also encodes the memory about the objects that the agent has already collected using a binary vector $s_{\mathrm{mem}} \in \{0, 1\}^{12}$. The $j^{\mathrm{th}}$ dimension encodes if the $j^{\mathrm{th}}$ object has been taken ($s_{\mathrm{mem},j} = 1$) or not ($s_{\mathrm{mem},j} = 0$). An additional constant term was added to the state to aid the function approximation. As a result, the state received by the agents is a column vector with $s = [s_{\mathrm{pos}}^\top, s_{\mathrm{mem}}^\top, 1]^\top \in \mathbb{R}^{113}$.

The features $\phi(s_t, a_t, s_{t+1}) \in \{0, 1\}^5$ in the environment describe the type of object that was collected by an agent during a step or if it reached the goal position. The first four feature dimensions encode binary the properties of a collected object and the last dimension if the goal area was reached. In total $|\Phi| = 6$ possible features exists: $\phi_1 = [0, 0, 0, 0, 0]^\top$ - standard observation, $\phi_2 = [1, 0, 1, 0, 0]^\top$ - collected an orange box, $\phi_3 = [1, 0, 0, 1, 0]^\top$ - collected an orange triangle, $\phi_4 = [0, 1, 1, 0, 0]^\top$ - collected a blue box, $\phi_5 = [0, 1, 0, 1, 0]^\top$ - collected a blue triangle, and $\phi_6 = [0, 0, 0, 0, 1]^\top$ - reached the goal area.

Two types of tasks were evaluated in this environment that have either 1) linear or 2) general reward functions. 300 tasks, i.e. reward functions, were sampled for each type. For linear tasks, the rewards $r = \phi^\top \mathbf{w}_i$ are defined by a linear combination of discrete features $\phi \in \mathbb{N}^5$ and a weight vector $\mathbf{w}_i \in \mathbb{R}^5$. The first four dimensions in $\mathbf{w}_i$ define the reward that the agent receives for collecting objects having specific properties, e.g. being blue or being a box. The weights for each of the four dimensions are randomly sampled from a uniform distribution: $\mathbf{w}_{k \in [1,2,3,4]} \sim \mathcal{U}(-1, 1)$ for each task. The final weight defines the reward for reaching the goal state which is $\mathbf{w}_5 = 1$ for each task. For training agents in general reward tasks, general reward functions $R_i$ for each task $M_i$ were sampled. These reward functions define for each of the four features $(\phi_2, \ldots, \phi_5)$ that represent the collection of a specific object type an individual reward. Their rewards were sampled from a uniform

distribution: $R_i(\phi_{k \in \{2,\dots,5\}}) \sim \mathcal{U}(-1, 1)$. The reward for collecting no object is $R_i(\phi_1) = 0$ and for reaching the goal area is $R_i(\phi_6) = 1$ for all tasks. Reward functions of this form can not be linearly decomposed in features and a weight vector.

## C.2 ALGORITHMS

This section introduces the details of each evaluated algorithm. First the common elements are discussed before introducing their specific implementations.

All agents experience the tasks $M \in \mathcal{M}$ of an environment sequentially. They are informed when a new task starts. All algorithms receive the features $\phi(s, a, s')$ of the environment. For the action selection and exploration, all agents use a $\epsilon$-greedy strategy. With probability $\epsilon \in [0, 1]$ the agent performs a random action. Otherwise it selects the action that maximizes the expected return.

As the state space ($s \in \mathbb{R}^{113}$) of the environments is high-dimensional and continuous, all agents use an approximation of their respective functions such as for the Q-function ($\tilde{Q}(s, a) \approx Q(s, a)$) or the $\xi$-function ($\tilde{\xi}(s, a, \phi) \approx \xi(s, a, \phi)$). We describe the general function approximation procedure on the example of $\xi$-functions. If not otherwise mentioned, all functions are approximated by a single linear mapping from the states to the function values. The parameters $\theta^\xi \in \mathbb{R}^{113 \times |\mathcal{A}| \times |\Phi|}$ of the mapping have independent components $\theta_{a,\phi}^\xi \in \mathbb{R}^{113}$ for each action $a \in \mathcal{A}$ and feature $\phi \in \Phi$:

$$\tilde{\xi}(s, a, \phi; \theta^\xi) = s^\top \theta_{a,\phi}^\xi \tag{35}$$

To learn $\tilde{\xi}$ we update the parameters $\theta^\xi$ using stochastic gradient descent following the gradients $\nabla_{\theta^\xi} \mathcal{L}_\xi(\theta^\xi)$ of the loss based on the $\xi$-learning update (13):

$$\forall \phi \in \Phi : \mathcal{L}_\xi(\theta^\xi) = \mathrm{E} \left\{ \left( p(\phi_t = \phi | s_t, a_t) + \gamma \tilde{\xi}(s_{t+1}, \bar{a}_{t+1}, \phi; \bar{\theta}^\xi) - \tilde{\xi}(s_t, a_t, \phi; \theta^\xi) \right)^2 \right\}$$
$$\text{with } \bar{a}_{t+1} = \underset{a}{\mathrm{argmax}} \sum_{\phi \in \Phi} R(\phi) \tilde{\xi}(s_{t+1}, a, \phi; \bar{\theta}^\xi) , \tag{36}$$

where $\bar{\theta}^\xi = \theta^\xi$ but $\bar{\theta}^\xi$ is treated as a constant for the purpose of calculating the gradients $\nabla_{\theta^\xi} \mathcal{L}_\xi(\theta^\xi)$. We used PyTorch[2] for the computation of gradients and its stochastic gradient decent procedure (SGD) for updating the parameters.

### C.2.1 QL

The Q-learning (QL) agent (Algorithm 1) represents standard Q-learning (Watkins and Dayan, 1992). The Q-function is approximated and updated using the following loss after each observed transition:

$$\mathcal{L}_Q(\theta^Q) = \mathrm{E} \left\{ \left( r(s_t, a_t, s_{t+1}) + \gamma \underset{a_{t+1}}{\max} \tilde{Q}(s_{t+1}, a_{t+1}; \bar{\theta}^Q) - \tilde{Q}(s_t, a_t; \theta^Q) \right)^2 \right\} \tag{37}$$

where $\bar{\theta}^Q = \theta^Q$ but $\bar{\theta}^Q$ is treated as a constant for the purpose of optimization, i.e no gradients flow through it. Following (Barreto et al., 2017) the parameters $\theta^Q$ are reinitialized for each new task.

### C.2.2 SFQL

The classical successor feature algorithm (SFQL) is based on a linear decomposition of rewards in features and reward weights (Barreto et al., 2017) (Algorithm 2). If the agent is learning the reward weights $\tilde{\mathbf{w}}_i$ for a task $M_i$ then they are randomly initialized at the beginning of a task. For the case of general reward functions and where the reward weights are given to the agents, the weights are learned to approximate a linear reward function before the task. See Section C.3 for a description of the training procedure. After each transition the weights are updated by minimizing the error between the predicted rewards $\phi(s_t, a_t, s_{t+1})^\top \tilde{\mathbf{w}}_i$ and the observed reward $r_t$:

$$\mathcal{L}_{\mathbf{w}_i}(\tilde{\mathbf{w}}_i) = \mathrm{E} \left\{ \left( r(s, a, s') - \phi(s_t, a_t, s_{t+1})^\top \tilde{\mathbf{w}}_i \right)^2 \right\} . \tag{38}$$

---

[2]PyTorch v1.4: https://pytorch.org

---

**Algorithm 1:** Q-learning (QL)

---

**Input :** exploration rate: $\epsilon$
    learning rate for the Q-function: $\alpha$

**for** $i \leftarrow 1$ **to** *num_tasks* **do**
  initialize $\tilde{Q}$: $\theta^Q \leftarrow$ small random initial values
  new_episode $\leftarrow$ true
  **for** $t \leftarrow 1$ **to** num_steps **do**
    **if** *new_episode* **then**
      new_episode $\leftarrow$ false
      $s_t \leftarrow$ initial state
    With probability $\epsilon$ select a random action $a_t$, otherwise $a_t \leftarrow \text{argmax}_a \tilde{Q}(s_t, a)$
    Take action $a_t$ and observe reward $r_t$ and next state $s_{t+1}$
    **if** $s_{t+1}$ *is a terminal state* **then**
      new_episode $\leftarrow$ true
      $\gamma_t \leftarrow 0$
    **else**
      $\gamma_t \leftarrow \gamma$
    $y \leftarrow r_t + \gamma_t \max_{a_{t+1}} \tilde{Q}(s_{t+1}, a_{t+1})$
    Update $\theta^Q$ using SGD($\alpha$) with $\mathcal{L}_Q = (y - \tilde{Q}(s_t, a_t))^2$
    $s_t \leftarrow s_{t+1}$

---

SFQL learns an approximated $\tilde{\psi}_i$-function for each task $M_i$. The parameters of the $\tilde{\psi}$-function for the first task $\theta_1^\psi$ are randomly initialized. For consecutive tasks, they are initialized by copying them from the previous task ($\theta_i^\psi \leftarrow \theta_{i-1}^\psi$). The $\tilde{\psi}_i$-function of the current task $M_i$ is updated after each observed transition with the loss based on (5):

$$\mathcal{L}_\psi(\theta_i^\psi) = \mathrm{E}\left\{ \left( \phi(s_t, a_t, s_{t+1}) + \gamma\tilde{\psi}_i(s_{t+1}, \bar{a}_{t+1}; \bar{\theta}_i^\psi) - \tilde{\psi}_i(s_t, a_t; \theta_i^\psi) \right)^2 \right\}$$

$$\text{with } \bar{a}_{t+1} = \text{argmax}_a \max_{k \in \{1,2,...,i\}} \tilde{\psi}_k(s_{t+1}, a; \bar{\theta}_k^\psi)^\top \tilde{\mathbf{w}}_i \,,$$

(39)

where $\bar{\theta}_i^\psi = \theta_i^\psi$ but $\bar{\theta}_i^\psi$ is treated as a constant for the purpose of optimization, i.e no gradients flow through it. Besides the current $\tilde{\psi}_i$-function, SFQL also updates the $\tilde{\psi}_c$-function which provided the GPI optimal action for the current transition: $c = \text{argmax}_{k \in \{1,2,...,i\}} \max_b \tilde{\psi}_k(s, b)^\top \tilde{\mathbf{w}}_i$. The update uses the same loss as for the update of the active $\tilde{\psi}_i$-function (39), but instead of using the GPI optimal action as next action, it uses the optimal action according to its own policy: $\bar{a}_{t+1} = \text{argmax}_a \tilde{\psi}_c(s_{t+1}, a)^\top \tilde{\mathbf{w}}_c$

### C.2.3  $\xi$-LEARNING

The $\xi$-learning agents (Algorithms 3, 4) allow to reevaluate policies in tasks with general reward functions. If the reward function is not given, an approximation $\tilde{R}_i$ of the reward function for each task $M_i$ is learned. The parameters for the approximation are randomly initialized at the beginning of each task. After each observed transition the approximation is updated according to the following loss:

$$\mathcal{L}_R(\theta_i^R) = \mathrm{E}\left\{ \left( r(s_t, a_t, s_{t+1}) - \tilde{R}_i(\phi(s_t, a_t, s_{t+1}); \theta_i^R) \right)^2 \right\}$$

(40)

In the case of tasks with linear reward functions the reward approximation becomes $\tilde{R}_i(\phi(s_t, a_t, s_{t+1}); \theta_i^R) = \phi(s_t, a_t, s_{t+1})^\top \theta_i^R$. Thus with $\theta_i^R = \tilde{\mathbf{w}}_i$ we recover the same procedure as for SFQL (38). For non-linear, general reward functions we represented $\tilde{R}$ with a neural network. The input of the network is the feature $\phi(s_t, a_t, s_{t+1})$. The network has one hidden layer with 10 neurons having ReLu activations. The output is a linear mapping to the scalar reward $r_t \in \mathbb{R}$.

All $\xi$-learning agents learn an approximation of the $\tilde{\xi}_i$-function for each task $M_i$. Analogous to SFQL, the parameters of the $\tilde{\xi}$-function for the first task $\theta_1^\xi$ are randomly initialized. For con-

---

**Algorithm 2:** Classical SF Q-learning (SFQL) (Barreto et al., 2017)

---

**Input :** exploration rate: $\epsilon$
       learning rate for $\psi$-functions: $\alpha$
       learning rate for reward weights $\mathbf{w}$: $\alpha_{\mathbf{w}}$
       features $\phi$
       optional: reward weights for tasks: $\{\tilde{\mathbf{w}}_1, \tilde{\mathbf{w}}_2, \ldots, \tilde{\mathbf{w}}_{\text{num\_tasks}}\}$

**for** $i \leftarrow 1$ **to** *num_tasks* **do**
    **if** $\tilde{\mathbf{w}}_i$ *not provided* **then** $\tilde{\mathbf{w}}_i \leftarrow$ small random initial values
    **if** $i = 1$ **then** initialize $\tilde{\psi}_i$: $\theta_i^{\psi} \leftarrow$ small random initial values **else** $\theta_i^{\psi} \leftarrow \theta_{i-1}^{\psi}$
    new_episode $\leftarrow$ true
    **for** $t \leftarrow 1$ **to** num_steps **do**
        **if** *new_episode* **then**
            new_episode $\leftarrow$ false
            $s_t \leftarrow$ initial state
        $c \leftarrow \operatorname{argmax}_{k \in \{1,2,\ldots,i\}} \max_a \tilde{\psi}_k(s_t, a)^{\top} \tilde{\mathbf{w}}_i$          // GPI optimal policy
        With probability $\epsilon$ select a random action $a_t$, otherwise $a_t \leftarrow \operatorname{argmax}_a \tilde{\psi}_c(s_t, a)^{\top} \tilde{\mathbf{w}}_i$
        Take action $a_t$ and observe reward $r_t$ and next state $s_{t+1}$
        Update $\tilde{\mathbf{w}}_i$ using SGD($\alpha_{\mathbf{w}}$) with $\mathcal{L}_{\mathbf{w}} = (r_t - \phi(s_t, a_t, s_{t+1})^{\top} \tilde{\mathbf{w}}_i)^2$
        **if** $s_{t+1}$ *is a terminal state* **then**
            new_episode $\leftarrow$ true
            $\gamma_t \leftarrow 0$
        **else**
            $\gamma_t \leftarrow \gamma$
        // GPI optimal next action for task $i$
        $\bar{a}_{t+1} \leftarrow \operatorname{argmax}_a \arg_{k \in \{1,2,\ldots,i\}} \tilde{\psi}_k(s_{t+1}, a)^{\top} \tilde{\mathbf{w}}_i$
        $y \leftarrow \phi(s_t, a_t, s_{t+1}) + \gamma_t \tilde{\psi}_i(s_{t+1}, \bar{a}_{t+1})$
        Update $\theta_i^{\psi}$ using SGD($\alpha$) with $\mathcal{L}_{\psi} = (y - \tilde{\psi}_i(s_t, a_t))^2$
        **if** $c \neq i$ **then**
            $\bar{a}_{t+1} \leftarrow \operatorname{argmax}_a \tilde{\psi}_c(s_{t+1}, a)^{\top} \tilde{\mathbf{w}}_c$   // optimal next action for task
            $c$
            $y \leftarrow \phi(s_t, a_t, s_{t+1}) + \gamma_t \tilde{\psi}_c(s_{t+1}, \bar{a}_{t+1})$
            Update $\theta_c^{\psi}$ using SGD($\alpha$) with $\mathcal{L}_{\psi} = (y - \tilde{\psi}_c(s_t, a_t))^2$
        $s_t \leftarrow s_{t+1}$

---

secutive tasks, they are initialized by copying them from the previous task ($\theta_i^{\xi} \leftarrow \theta_{i-1}^{\xi}$). The $\tilde{\xi}_i$-function of the current task $M_i$ is updated after each observed transition with the loss given in (46). The $\xi$-learning agents differ in their setting for $p(\phi_t = \phi | s_t, a_t)$ in the updates which is described in the upcoming sections. Besides the current $\tilde{\xi}_i$-function, the $\xi$-learning agents also update the $\tilde{\xi}_c$-function which provided the GPI optimal action for the current transition: $c = \operatorname{argmax}_{k \in \{1,2,\ldots,i\}} \max_{a_t} \sum_{\phi \in \Phi} \tilde{\xi}_k(s_t, a_t, \phi) \tilde{R}_i(\phi)$. The update uses the same loss as for the update of the active $\tilde{\xi}_i$-function (46), but instead of using the GPI optimal action as next action, it uses the optimal action according to its own policy: $\bar{a}_{t+1} = \max_a \sum_{\phi \in \Phi} \tilde{\xi}_c(s_{t+1}, a, \phi) \tilde{R}_c(\phi)$.

**MF $\xi$-learning:** The model-free $\xi$-learning agent (Algorithm 3) uses a stochastic update for the $\tilde{\xi}$-functions. Given a transition, we set $p(\phi_t = \phi | s_t, a_t) \equiv 1$ for the observed feature $\phi = \phi(s_t, a_t, s_{t+1})$ and $p(\phi_t = \phi | s_t, a_t) \equiv 0$ for all other features $\phi \neq \phi(s_t, a_t, s_{t+1})$.

**MB $\xi$-learning:** The one-step SF model-based $\xi$-learning agent (Algorithm 4) uses an approximated model $\tilde{p}$ to predict $p(\phi_t = \phi | s_t, a_t)$ to reduce the variance of the $\xi$-function update. The model is by

a linear mapping for each action. It uses a softmax activation to produce a valid distribution over $\Phi$:

$$\tilde{p}(s, a, \phi; \theta^p) = \frac{\exp(s^\top \theta^p_{a,\phi})}{\sum_{\phi' \in \Phi} \exp(s^\top \theta^p_{a,\phi'})} \tag{41}$$

where $\theta^p_{a,\phi} \in \mathbb{R}^{113}$. As $\tilde{p}$ is valid for each task in $\mathcal{M}$, its weights $\theta^p$ are only randomly initialized at the beginning of the first task. For each observed transition, the model is updated using the following loss:

$$\forall \phi \in \Phi : \ \mathcal{L}_p(\theta^p_i) = \mathrm{E}\left\{ (p(\phi_t = \phi | s_t, a_t) - \tilde{p}(s_t, a_t, \phi; \theta^p_i))^2 \right\} , \tag{42}$$

where we set $p(\phi_t = \phi | s_t, a_t) \equiv 1$ for the observed feature $\phi = \phi(s_t, a_t, s_{t+1})$ and $p(\phi_t = \phi | s_t, a_t) \equiv 0$ for all other features $\phi \neq \phi(s_t, a_t, s_{t+1})$.

### C.3 EXPERIMENTAL PROCEDURE

All agents were evaluated in both task types (linear or general reward function) on 300 tasks. The agents experienced the tasks sequentially, each for 20.000 steps. The agents had knowledge when a task change happened. Each agent was evaluated for 10 repetitions to measure their average performance. Each repetition used a different random seed that impacted the following elements: a) the sampling of the tasks, b) the random initialization of function approximator parameters, c) the stochastic behavior of the environments when taking steps, and d) the $\epsilon$-greedy action selection of the agents. The tasks, i.e. the reward functions, were different between the repetitions of a particular agent, but identical to the same repetition of a different agent. Thus, all algorithms were evaluated over the same tasks.

The SF agents (SFQL, $\xi$-learning) were evaluated under two conditions. First, that they have to learn the reward weights or the reward function online during the training (indicated by (O) in figures). Second, that the reward weights or the reward function is given to them. As the SFQL does not support general reward functions, it is not possible to provide the SFQL agent with the reward function in the second condition. As a solution, before the agent was trained on a new task $\mathcal{M}_i$, a linear model of the reward $R_i(\phi) = \phi^\top \tilde{w}_i$ was fitted. The initial approximation $\tilde{w}_i$ was randomly initialized and then fitted for 10.000 iterations using a gradient descent procedure based on the absolute mean error (L1 norm):

$$\Delta \tilde{w}_i = \eta \frac{1}{|\Phi|} \sum_{\phi \in \Phi} R_i(\phi) - \phi^\top \tilde{w}_i , \tag{43}$$

with a learning rate of $\eta = 1.0$ that yielded the best results tested over several learning rates.

**Hyperparameters:** The hyperparameters of the algorithms were set to the same values as in (Barreto et al., 2017). A grid search over the learning rates of all algorithms was performed. Each learning rate was evaluated for three different settings which are listed in Table 1. If algorithms had several learning rates, then all possible combinations were evaluated. This resulted in a different number of evaluations per algorithm and condition: QL - 3, SFQL (O) - 9, MF Xi (O) - 9, MB Xi (O) - 27, SFQL - 3, MF Xi - 3, MB Xi - 9. In total, 63 parameter combinations were evaluated. The reported performances in the figures are for the parameter combination that resulted in the highest cumulative total reward averaged over all 10 repetitions in the respective environment. Please note, the learning rates $\alpha$ and $\alpha_{\mathbf{w}}$ are set to half of the rates defined in (Barreto et al., 2017). This is necessary due to the differences in calculating the loss and the gradients in the current paper. We use mean squared error loss formulations, whereas (Barreto et al., 2017) uses absolute error losses. The probability for random actions of the $\epsilon$-Greedy action selection was set to $\epsilon = 0.15$ and the discount rate to $\gamma = 0.95$. The initial weights $\theta$ for the function approximators were randomly sampled from a standard distribution with $\theta_{\text{init}} \sim \mathcal{N}(\mu = 0, \sigma = 0.01)$.

**Computational Resources:** Experiments were conducted on a cluster with a variety of node types (Xeon SKL Gold 6130 with 2.10GHz, Xeon SKL Gold 5218 with 2.30GHz, Xeon SKL Gold 6126 with 2.60GHz, Xeon SKL Gold 6244 with 3.60GHz, each with 192 GB Ram, no GPU). The time for evaluating one repetition of a certain parameter combination over the 300 tasks depended on the algorithm and the task type. Linear reward function tasks: QL $\approx 1h$, SFQL (O) $\approx 4h$, MF $\xi$ (O) $\approx 42h$, MB $\xi$ (O) $\approx 43h$, SFQL $\approx 4h$, MF $\xi \approx 15h$, and MB $\xi \approx 16h$. General reward function

Table 1: Evaluated Learning Rates in the Object Collection Environment

| Parameter | Description | Values |
|-----------|-------------|--------|
| $\alpha$ | Learning rate of the Q, $\psi$, and $\xi$-function | $\{0.0025, 0.005, 0.025\}$ |
| $\alpha_{\mathbf{w}}, \alpha_R$ | Learning rate of the reward weights or the reward model | $\{0.025, 0.05, 0.075\}$ |
| $\beta$ | Learning rate of the One-Step SF Model | $\{0.2, 0.4, 0.6\}$ |

tasks: QL $\approx 1h$, SFQL (O) $\approx 4h$, MF $\xi$ (O) $\approx 68h$, MB $\xi$ (O) $\approx 67h$, SFQL $\approx 5h$, MF $\xi \approx 14h$, and MB $\xi \approx 18h$. Please note, the reported times do not represent well the computational complexity of the algorithms, as the algorithms were not optimized for speed, and some use different software packages (numpy or pytorch) for their individual computations.

### C.4 Effect of Increasing Non-linearity in General Reward Task

We further studied the effect of general reward functions on the performance of classical SF compared to $\xi$-learning (Fig. 1 - c). We evaluated the agents in tasks with different levels of difficulty in relation to how well their reward functions can be approximated by a linear model. Seven difficulty levels have been evaluated. For each level, the agents were trained sequentially on 300 tasks as for the experiments with general reward functions. The reward functions for each level were sampled with the following procedure. Several general reward functions were randomly sampled as previously described. For each reward function, a linear model of a reward weight vector $\tilde{\mathbf{w}}$ was fitted using the same gradient descent procedure as in Eq. 43. The final average absolute model error after 10.000 iterations was measured. Each of the seven difficulty levels defines a range of model errors its tasks have with the following increasing ranges: $\{[0.0, 0.25], [0.25, 0.5], \ldots, [1.5, 1.75]\}$. For each difficulty level, 300 reward functions were selected that yield a linear model are in the respective range of the level.

Q-Learning, SFQL, and MF Xi-learning were each trained on 300 tasks, i.e. reward functions, on each difficulty level. As hyperparameters were the best performing parameters from the previous general reward task experiments used. We measured the ratio between the total return over 300 tasks of QL and MF Xi-learning (QL/MF Xi), and SFQL and MF Xi-learning (SFQL/MF Xi). Fig. 1 - c shows the results, using as x-axis the mean average absolute model error defined by the bracket of each difficulty level. The results show that the relative performance of SFQL compared to MF Xi reduces with higher non-linearity of the reward functions. For reward functions that are nearly linear (mean error of 0.125), both have a similar performance. Whereas, for reward functions that are difficult to model with a linear relation (mean error of 1.625) SFQL reaches only less than 50% of the performance of MF Xi-learning.

---

**Algorithm 3:** Model-free $\xi$-learning (MF $\xi$)

---

**Input :** exploration rate: $\epsilon$
learning rate for $\xi$-functions: $\alpha$
learning rate for reward models $R$: $\alpha_R$
features $\phi$
optional: reward functions for tasks: $\{\tilde{R}_1, \tilde{R}_2, \ldots, \tilde{R}_{\text{num\_tasks}}\}$

**for** $i \leftarrow 1$ **to** *num\_tasks* **do**
    **if** $\tilde{R}_i$ *not provided* **then** initialize $\tilde{R}_i$: $\theta_i^R \leftarrow$ small random initial values
    **if** $i = 1$ **then** initialize $\tilde{\xi}_i$: $\theta_i^\xi \leftarrow$ small random initial values **else** $\theta_i^\xi \leftarrow \theta_{i-1}^\xi$
    new\_episode $\leftarrow$ true
    **for** $t \leftarrow 1$ **to** num\_steps **do**
        **if** *new\_episode* **then**
            new\_episode $\leftarrow$ false
            $s_t \leftarrow$ initial state
        $c \leftarrow \mathrm{argmax}_{k \in \{1,2,\ldots,i\}} \max_a \sum_\phi \tilde{\xi}_k(s_t, a, \phi) \tilde{R}_i(\phi)$   `// GPI optimal policy`
        With probability $\epsilon$ select a random action $a_t$, otherwise
            $a_t \leftarrow \mathrm{argmax}_a \sum_\phi \tilde{\xi}_c(s_t, a, \phi) \tilde{R}_i(\phi)$
        Take action $a_t$ and observe reward $r_t$ and next state $s_{t+1}$
        **if** $\tilde{R}_i$ *not provided* **then**
            Update $\theta_i^R$ using SGD($\alpha_R$) with $\mathcal{L}_R = (r_t - \tilde{R}_i(\phi(s_t, a_t, s_{t+1})))^2$
        **if** $s_{t+1}$ *is a terminal state* **then**
            new\_episode $\leftarrow$ true
            $\gamma_t \leftarrow 0$
        **else**
            $\gamma_t \leftarrow \gamma$
        `// GPI optimal next action for task i`
        $\bar{a}_{t+1} \leftarrow \mathrm{argmax}_a \arg_{k \in \{1,2,\ldots,i\}} \sum_\phi \tilde{\xi}_k(s_{t+1}, a, \phi) \tilde{R}_i(\phi)$
        **foreach** $\phi \in \Phi$ **do**
            **if** $\phi = \phi(s_t, a_t, s_{t+1})$ **then** $y_\phi \leftarrow 1 + \gamma_t \tilde{\xi}_i(s_{t+1}, \bar{a}_{t+1}, \phi)$
            **else** $y_\phi \leftarrow \gamma_t \tilde{\xi}_i(s_{t+1}, \bar{a}_{t+1}, \phi)$
        Update $\theta_i^\xi$ using SGD($\alpha$) with $\mathcal{L}_\xi = \sum_\phi (y_\phi - \tilde{\xi}_i(s_t, a_t, \phi))^2$
        **if** $c \neq i$ **then**
            `// optimal next action for task c`
            $\bar{a}_{t+1} \leftarrow \mathrm{argmax}_a \sum_\phi \tilde{\xi}_c(s_{t+1}, a, \phi) \tilde{R}_c(\phi)$
            **foreach** $\phi \in \Phi$ **do**
                **if** $\phi = \phi(s_t, a_t, s_{t+1})$ **then** $y_\phi \leftarrow 1 + \gamma_t \tilde{\xi}_c(s_{t+1}, \bar{a}_{t+1}, \phi)$
                **else** $y_\phi \leftarrow \gamma_t \tilde{\xi}_c(s_{t+1}, \bar{a}_{t+1}, \phi)$
            Update $\theta_c^\xi$ using SGD($\alpha$) with $\mathcal{L}_\xi = \sum_\phi (y_\phi - \tilde{\xi}_c(s_t, a_t, \phi))^2$
        $s_t \leftarrow s_{t+1}$

---

---

**Algorithm 4:** One Step SF-Model $\xi$-learning (MB $\xi$)

---

**Input :** exploration rate: $\epsilon$

      learning rate for $\xi$-functions: $\alpha$

      learning rate for reward models $R$: $\alpha_R$

      learning rate for the one-step SF model $\tilde{p}$: $\beta$

      features $\phi$

      optional: reward functions for tasks: $\{\tilde{R}_1, \tilde{R}_2, \ldots, \tilde{R}_{\text{num\_tasks}}\}$

initialize $\tilde{p}$: $\theta^p \leftarrow$ small random initial values

**for** $i \leftarrow 1$ **to** *num_tasks* **do**

    **if** $\tilde{R}_i$ *not provided* **then** initialize $\tilde{R}_i$: $\theta_i^R \leftarrow$ small random initial values

    **if** $i = 1$ **then** initialize $\tilde{\xi}_i$: $\theta_i^\xi \leftarrow$ small random initial values **else** $\theta_i^\xi \leftarrow \theta_{i-1}^\xi$

    new_episode $\leftarrow$ true

    **for** $t \leftarrow 1$ **to** num_steps **do**

        **if** *new_episode* **then**

            new_episode $\leftarrow$ false

            $s_t \leftarrow$ initial state

        $c \leftarrow \text{argmax}_{k \in \{1,2,\ldots,i\}} \max_a \sum_\phi \tilde{\xi}_k(s_t, a, \phi)\tilde{R}_i(\phi)$    `// GPI optimal policy`

        With probability $\epsilon$ select a random action $a_t$, otherwise

            $a_t \leftarrow \text{argmax}_a \sum_\phi \tilde{\xi}_c(s_t, a, \phi)\tilde{R}_i(\phi)$

        Take action $a_t$ and observe reward $r_t$ and next state $s_{t+1}$

        **if** $\tilde{R}_i$ *not provided* **then**

            Update $\theta_i^R$ using SGD($\alpha_R$) with $\mathcal{L}_R = (r_t - \tilde{R}_i(\phi(s_t, a_t, s_{t+1})))^2$

        **foreach** $\phi \in \Phi$ **do**

            **if** $\phi = \phi(s_t, a_t, s_{t+1})$ **then** $y_\phi \leftarrow 1$ **else** $y_\phi \leftarrow 0$

        Update $\theta^p$ using SGD($\beta$) with $\mathcal{L}_p = \sum_\phi (y_\phi - \tilde{p}(s_t, a_t, \phi))^2$

        **if** $s_{t+1}$ *is a terminal state* **then**

            new_episode $\leftarrow$ true

            $\gamma_t \leftarrow 0$

        **else**

            $\gamma_t \leftarrow \gamma$

        `// GPI optimal next action for task` $i$

        $\bar{a}_{t+1} \leftarrow \text{argmax}_a \arg_{k \in \{1,2,\ldots,i\}} \sum_\phi \tilde{\xi}_k(s_{t+1}, a, \phi)\tilde{R}_i(\phi)$

        **foreach** $\phi \in \Phi$ **do**

            $y_\phi \leftarrow \tilde{p}(s_t, a_t, \phi) + \gamma_t \tilde{\xi}_i(s_{t+1}, \bar{a}_{t+1}, \phi)$

        Update $\theta_i^\xi$ using SGD($\alpha$) with $\mathcal{L}_\xi = \sum_\phi (y_\phi - \tilde{\xi}_i(s_t, a_t, \phi))^2$

        **if** $c \neq i$ **then**

            `// optimal next action for task` $c$

            $\bar{a}_{t+1} \leftarrow \text{argmax}_a \sum_\phi \tilde{\xi}_c(s_{t+1}, a, \phi)\tilde{R}_c(\phi)$

            **foreach** $\phi \in \Phi$ **do**

                 $y_\phi \leftarrow \tilde{p}(s_t, a_t, \phi) + \gamma_t \tilde{\xi}_c(s_{t+1}, \bar{a}_{t+1}, \phi)$

            Update $\theta_c^\xi$ using SGD($\alpha$) with $\mathcal{L}_\xi = \sum_\phi (y_\phi - \tilde{\xi}_c(s_t, a_t, \phi))^2$

        $s_t \leftarrow s_{t+1}$

---

## D    EXPERIMENTAL DETAILS: RACER ENVIRONMENT

This section extends the brief introduction to the racer environment (Fig. 2 - a) given in Section 4.2.

### D.1    ENVIRONMENT

The environment is a continuous two-dimensional area in which the agent drives similar to a car. The position of the agent is a point in the 2D space: $p = (x, y) \in [0, 1]^2$. Moreover, the agent has an orientation which it faces: $\theta \in [-\pi, \pi 1]$. The action space of the agent consists of three movement directions: $A = \{$right, straight, left $\}$. Each action changes the position of the agent depending on its current position and orientation. The action *straight* changes the agent's position by $0.075$ towards its orientation $\theta$. The action *right* changes the orientation of the agent to $\theta + \frac{1}{7}\pi$ and its position $0.06$ towards this new direction, whereas *left* the direction to $\theta - \frac{1}{7}\pi$ changes. The environment is stochastic by adding Gaussian noise with $\sigma = 0.005$ to the final position $x, y$, and orientation $\theta$. If the agent drives outside the area $(x, y) \in [0, 1]^2$, then it reappears on the other opposite side. The environment resembles therefore a torus (or donut). As a consequence, distances $d(p_x, p_y)$ are also measure in this space, so that the positions $p_x = (0.1, 0.5)$ and $p_y = (0.9, 0.5)$ have not a distance of $0.8$ but $d(p_x, p_y) = 0.2$. The environment has 3 markers at the positions $m_1 = (0.25, 0.75)$, $m_2 = (0.75, 0.25)$, and $m_3 = (0.75, 0.6)$. The features measure the distance of the agent to each marker: $\phi \in \mathbb{R}^3$ with $\phi_k = d(p, m_k)$. Each feature dimensions is normalized to be $\phi_k \in [0, 1]$. At the beginning of an episode the agent is randomly placed and oriented in environment. An episode ends after 200 time steps.

The state space of the agents is similarly constructed as for the object collection environment. The agent's position is encoded with a $10 \times 10$ radial basis functions $s_{\text{pos}} \in \mathbb{R}^{100}$ as defined in 34. In difference, that the distances are measure according to the torus shape. A similar radial basis function approach is also used to encode the orientation $s_{\text{ori}} \in \mathbb{R}^{20}$ of the agent using 20 equally distributed gaussian centers in $[-\pi, \pi]$ and $\sigma = \frac{1}{5}\pi$. Please note, $\pi$ and $-\pi$ are also connected in this space, i.e. $d(\pi, -\pi) = 0$. The combination of the position and orientation of the agent is the final state: $s = [s_{\text{pos}}^\top, s_{\text{ori}}^\top]^\top \in \mathbb{R}^{120}$.

The reward functions define preferred positions in the environment based on the features, i.e. the distance of the agent to the markers. A preference function $r_k$ exists for each distance. The functions are composed of a maximization over $m$ Gaussian components that evaluate the agents distance:

$$R(\phi) = \sum_{k=1}^{3} r_k(\phi_k) \quad \text{with} \quad r_i = \frac{1}{3} \max \left\{ \exp \left( -\frac{(\phi_k - \mu_j)^2}{\sigma_j} \right) \right\}_{j=1}^{m}. \tag{44}$$

Reward functions are randomly generated by sampling the number of Gaussian components $m$ to be 1 or 2. The properties of each component are sampled according to $\mu_j \sim \mathcal{U}(0.0, 0.7)$, and $\sigma_j \sim \mathcal{U}(0.001, 0.01)$. Fig. 2 - a illustrates one such randomly sampled reward function where dark areas represent locations with high rewards.

### D.2    ALGORITHMS

We evaluated Q-learning, SFQL (O), SFQL, and MF $\xi$-learning (see Section C.2 for their full description) in the racer environment. In difference to their implementation for the object collection environment, they used a different neural network architecture to approximate their respective value functions.

#### D.2.1    QL

Q-learning uses a fully connected feedforward network with bias and a ReLU activation for hidden layers. It has 2 hidden layers with 20 neurons each.

#### D.2.2    SFQL

Q-learning uses a feedforward network with bias and a ReLU activation for hidden layers. It has for each of the three feature dimensions a separate fully connected subnetwork. Each subnetwork has 2 hidden layers with 20 neurons each.

### D.2.3 CONTINUOUS MODEL-FREE $\xi$-LEARNING

The racer environment has continuous features $\phi \in \mathbb{R}^3$. Therefore, the MF $\xi$-learning procedure (Alg. 3) can not be directly applied as it is designed for discrete feature spaces. We introduce here a MF $\xi$-learning procedure for continuous feature spaces (CMF $\xi$-learning). It is a feature dimension independent, and discretized version of $\xi$-learning. As the reward functions (44) are a sum over the individual feature dimensions, the Q-value can be computed as:

$$Q^\pi(s,a) = \int_\Phi R(\phi)\xi^\pi(s,a,\phi)\mathrm{d}\phi = \sum_k \int_{\Phi_k} r_k(\phi_k)\xi_k^\pi(s,a,\phi_k)\mathrm{d}\phi_k , \tag{45}$$

where $\Phi_k$ is the feature space for each feature dimension which is $\Phi_k = [0,1]$ in the racer environment. $\xi_k^\pi$ is a $\xi$-function for the feature dimension $k$. (45) shows that the $\xi$-function can be independently represented over each individual feature dimension $\phi_k$, instead of over the full features $\phi$. This reduces the complexity of the approximation.

Moreover, we introduce a discretization of the $\xi$-function that discretizes the space of each feature dimension $k$ in $U = 11$ bins with the centers:

$$X_k = \left\{ \phi_k^{\min} + j\Delta\phi_k : 0 < j < U \right\}, \quad \text{with} \ \ \Delta\phi_k := \frac{\phi_k^{max} - \phi_k^{min}}{U-1} ,$$

where $\Delta\phi_k$ is the distance between the centers, and $\phi_k^{min} = 0.0$ is the lowest center, and $\phi_k^{max} = 1.0$ the largest center. Given this discretization and the decomposition of the Q-function according to (45), the Q-values can be computed by:

$$Q^\pi(s,a) = \sum_k \sum_{x \in X_k} R(x)\xi^\pi(s,a,x) .$$

Alg. 5 lists the complete CMF $\xi$-learning procedure with the update steps for the $\xi$-functions. Similar to the SFQL agent, the CMF Xi uses a feedforward network with bias and a ReLU activation for hidden layers. It has for each of the three feature dimensions a separate fully connected subnetwork. Each subnetwork has 2 hidden layers with 20 neurons each. The discretized outputs per feature dimension share the last hidden layer per subnetwork.

The $\xi$-function is updated according to the following procedure. Instead of providing a discrete learning signal to the model, we encode the observed feature using continuous activation functions around each bin center. Given the $j$'th bin center of dimension $k$, $x_{k,j}$, its value is encoded to be 1.0 if the feature value of this dimension aligns with the center ($\phi_k = x_{k,j}$). Otherwise, the encoding for the bin decreases linearly based on the distance between the bin center and the value ($|x_{k,j} - \phi_k|$) and reaches 0 if the value is equal to a neighboring bin center, i.e. has a distance $\geq \Delta\phi_k$. We represent this encoding for each feature dimension $k$ by $\mathbf{u}_k \in (0,1)^U$ with:

$$\forall_{k \in \{1,2,3\}} : \forall_{0 < j < U} : u_{k,j} = \max\left(0, \frac{(1 - |x_{k,j} - \phi_k|)}{\Delta\phi_k}\right) .$$

To learn $\tilde{\xi}$ we update the parameters $\theta^\xi$ using stochastic gradient descent following the gradients $\nabla_{\theta^\xi}\mathcal{L}_\xi(\theta^\xi)$ of the loss based on the $\xi$-learning update (13):

$$\forall\phi \in \Phi : \ \mathcal{L}_\xi(\theta^\xi) = \mathrm{E}\left\{\frac{1}{n}\sum_{k=1}^3 \left(\mathbf{u}_k + \gamma\tilde{\xi}_k(s_{t+1}, \bar{a}_{t+1}; \bar{\theta}^\xi) - \tilde{\xi}_k(s_t, a_t; \theta^\xi)\right)^2\right\}$$
$$\text{with} \ \bar{a}_{t+1} = \underset{a}{\mathrm{argmax}} \sum_k \sum_{x \in X_k} R(x)\tilde{\xi}(s_{t+1}, a, \phi; \bar{\theta}^\xi) , \tag{46}$$

where $n = 3$ is the number of feature dimensions and $\tilde{\xi}_k$ is the vector of the $U$ discretized $\xi$-values for dimension $k$.

### D.3 EXPERIMENTAL PROCEDURE

All agents were evaluated on 37 tasks. The agents experienced the tasks sequentially, each for 1000 episodes ($200,000$ steps per task). The agents had knowledge when a task change happened.

Each agent was evaluated for 10 repetitions to measure their average performance. Each repetition used a different random seed that impacted the following elements: a) the sampling of the tasks, b) the random initialization of function approximator parameters, c) the stochastic behavior of the environments when taking steps, and d) the $\epsilon$-greedy action selection of the agents. The tasks, i.e. the reward functions, were different between the repetitions of a particular agent, but identical to the same repetition of a different agent. Thus, all algorithms were evaluated over the same tasks.

SFQL was evaluated under two conditions. First, by learning the reward weights online during the training (indicated by (O) in figures). Second, the reward weights were trained with the iterative gradient decent method in (43). The weights were trained for $10,000$ iterations with an learning rate of $1.0$. At each iteration, $50$ random points in the task were sampled and their features and rewards are used for the training step.

**Hyperparameters**    A grid search over the learning rates of all algorithms was performed. Each learning rate was evaluated for three different settings which are listed in Table 2. If algorithms had several learning rates, then all possible combinations were evaluated. This resulted in a different number of evaluations per algorithm and condition: QL - 4, SFQL (O) - 12, SFQL - 4, CMF Xi 4. In total, 24 parameter combinations were evaluated. The reported performances in the figures are for the parameter combination that resulted in the highest cumulative total reward averaged over all 10 repetitions in the respective environment. The probability for random actions of the $\epsilon$-Greedy action selection was set to $\epsilon = 0.15$ and the discount rate to $\gamma = 0.9$. The initial weights and biases $\theta$ for the function approximators were initialized according to an uniform distribution with $\theta_i \sim \mathcal{U}(-\sqrt{k}, \sqrt{k})$, where $k = \frac{1}{\text{in\_features}}$.

Table 2: Evaluated Learning Rates in the Racer Environment

| Parameter | Description | Values |
|-----------|-------------|--------|
| $\alpha$ | Learning rate of the Q, $\psi$, and $\xi$-function | $\{0.0025, 0.005, 0.025, 0.5\}$ |
| $\alpha_{\mathbf{w}}$ | Learning rate of the reward weights | $\{0.025, 0.05, 0.075\}$ |

**Computational Resources:**    Experiments were conducted on the same cluster as for the object collection environment experiments. The time for evaluating one repetition of a certain parameter combination over the 37 tasks depended on the algorithm: QL $\approx 9h$, SFQL (O) $\approx 70h$, SFQL $\xi$ $\approx 73h$, and CMF $\xi \approx 88h$. Please note, the reported times do not represent well the computational complexity of the algorithms, as the algorithms were not optimized for speed, and some use different software packages (numpy or pytorch) for their individual computations.

---

**Algorithm 5:** Model-free $\xi$-learning for Continuous Features (CMF $\xi$)

---

**Input :** exploration rate: $\epsilon$
learning rate for $\xi$-functions: $\alpha$
learning rate for reward models $R$: $\alpha_R$
features $\phi \in \mathbb{R}^n$
components of reward functions for tasks: $\{R_1 = \{r_1^1, r_2^1, ..., r_n^1\}, R_2, \ldots, R_{\text{num\_tasks}}\}$
discretization parameters: $X, \Delta\phi$

**for** $i \leftarrow 1$ **to** *num_tasks* **do**
 **if** $i = 1$ **then**
  $\forall_{k \in \{1,...,n\}}$: initialize $\tilde{\xi}_k^i$: $\theta_{i,k}^\xi \leftarrow$ small random values
 **else**
  $\forall_{k \in \{1,...,n\}}$: $\theta_{i,k}^\xi \leftarrow \theta_{i-1,k}^\xi$
 new_episode $\leftarrow$ true
 **for** $t \leftarrow 1$ **to** num_steps **do**
  **if** *new_episode* **then**
   new_episode $\leftarrow$ false
   $s_t \leftarrow$ initial state
  $c \leftarrow \text{argmax}_{j \in \{1,2,...,i\}} \max_a \sum_{k=1}^n \sum_{x \in X_k} \tilde{\xi}_k^j(s_t, a, x) r_k^i(x)$   // GPI policy
  With probability $\epsilon$ select a random action $a_t$, otherwise
   $a_t \leftarrow \text{argmax}_a \sum_{k=1}^n \sum_{x \in X_k} \tilde{\xi}_k^j(s_t, a, x) r_k^i(x)$
  Take action $a_t$ and observe reward $r_t$ and next state $s_{t+1}$
  **if** $s_{t+1}$ *is a terminal state* **then**
   new_episode $\leftarrow$ true
   $\gamma_t \leftarrow 0$
  **else**
   $\gamma_t \leftarrow \gamma$
  // GPI optimal next action for task $i$
  $\bar{a}_{t+1} \leftarrow \text{argmax}_a \arg_{j \in \{1,2,...,i\}} \sum_{k=1}^n \sum_{x \in X_k} \tilde{\xi}_k^j(s_t, a, x) r_k^i(x)$
  $\phi_t \leftarrow \phi(s_t, a_t, s_{t+1})$
  **for** $k \leftarrow 1$ **to** $n$ **do**
   **foreach** $x \in X_k$ **do**
    $y_{k,x} \leftarrow \max\left(0, 1 - \frac{|x - \phi_{t,k}|}{\Delta\phi}\right) + \gamma_t \tilde{\xi}_k^i(s_{t+1}, \bar{a}_{t+1}, x)$
  Update $\theta_i^\xi$ using SGD($\alpha$) with $\mathcal{L}_\xi = \sum_{k=1}^n \sum_{x \in X_k} (y_{k,x} - \tilde{\xi}_k^i(s_t, a_t, x))^2$
  **if** $c \neq i$ **then**
   // optimal next action for task $c$
   $\bar{a}_{t+1} \leftarrow \text{argmax}_a \sum_{k=1}^n \sum_{x \in X_k} \tilde{\xi}_k^c(s_t, a, x) r_k^c(x)$
   **for** $k \leftarrow 1$ **to** $n$ **do**
    **foreach** $x \in X_k$ **do**
     $y_{k,x} \leftarrow \max\left(0, 1 - \frac{|x - \phi_{t,k}|}{\Delta\phi}\right) + \gamma_t \tilde{\xi}_k^c(s_{t+1}, \bar{a}_{t+1}, x)$
   Update $\theta_c^\xi$ using SGD($\alpha$) with $\mathcal{L}_\xi = \sum_{k=1}^n \sum_{x \in X_k} (y_{k,x} - \tilde{\xi}_k^c(s_t, a_t, x))^2$
  $s_t \leftarrow s_{t+1}$

---

# E    ADDITIONAL EXPERIMENTAL RESULTS

This section reports additional results and experiments:

1. Report of the total return and the statistical significance of differences between agents for all experiments

2. Evaluation of the agents in the original object collection task by Barreto et al. (2017)

## E.1    OBJECT COLLECTION TASK BY BARRETO ET AL. (2017)

We additionally evaluated all agents in the original object collection task by Barreto et al. (2017).

**Environment:**    The environment differs to the modified object collection task (Section. C) only in terms of the objects and features. The environment has 3 object types: orange, blue, and pink (Fig. 3). The feature encode if the agent has collected one of these object types or if it reached the goal area. The first three dimensions of the features $\phi(s_t, a_t, s_{t+1}) \in \{0, 1\}^4$ encode which object type is collected. The last dimension encodes if the goal area was reached. In total $|\Phi| = 5$ possible features exists: $\phi_1 = [0, 0, 0, 0]^\top$ - standard observation, $\phi_2 = [1, 0, 0, 0]^\top$ - collected an orange object, $\phi_3 = [0, 1, 0, 0]^\top$ - collected a blue object, $\phi_4 = [0, 0, 1, 0]^\top$ - collected a pink object, and $\phi_5 = [0, 0, 0, 1]^\top$ - reached the goal area.

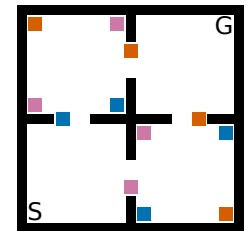

Figure 3:    Object collection environment from (Barreto et al., 2017) with 3 object types: orange, blue, pink.

The rewards $r = \phi^\top \mathbf{w}_i$ are defined by a linear combination of discrete features $\phi \in \mathbb{N}^4$ and a weight vector $\mathbf{w} \in \mathbb{R}^4$. The first three dimensions in $\mathbf{w}$ define the reward that the agent receives for collecting one of the object types. The final weight defines the reward for reaching the goal state which is $\mathbf{w}_4 = 1$ for each task. All agents were trained in on 300 randomly generated linear reward functions with the same experimental procedure as described in Section. C. For each task the reward weights for the 3 objects are randomly sampled from a uniform distribution: $\mathbf{w}_{k \in \{1,2,3\}} \sim \mathcal{U}(-1, 1)$.

**Results:**    The results (Fig. 4) follow closely the results from the modified object collection task (Fig. 1 - b, and 5 - a). MF $\xi$ reaches the highest performance outperforming SFQL in terms of learning speed and asymptotic performance. It is followed by MB $\xi$ and SFQL which show no statistical significant difference between each other in their final performance. Nonetheless, MB $\Xi$ has a higher learning speed during the initial 40 tasks. The results for the agents that learn the reward weights online (SFQL (O), MF $\xi$ (O), and MB $\xi$ (O)) follow the same trend with MF $\xi$ (O) outperforming SFQL (O) slightly. Nonetheless, the $\xi$-agents have a much stronger learning speed during the initial 70 tasks compared to SFQL (O), due to the errors in the approximation of the weight vectors, especially at the beginning of a new task. All agents can clearly outperform standard Q-learning.

## E.2    TOTAL RETURN IN TRANSFER LEARNING EXPERIMENTS AND STATISTICAL SIGNIFICANT DIFFERENCES

Fig. 5 shows for each of the transfer learning experiments in the object collection and the racer environment the total return that each agent accumulated over all tasks. Each dot besides the boxplot shows the total return for each of the 10 repetitions. The box ranges from the upper to the lower quartile. The whiskers represent the upper and lower fence. The mean and standard deviation are indicated by the dashed line and the median by the solid line. The tables in Fig. 5 report the p-value of pairwise Mann–Whitney U test. A significant different total return can be expected if $p < 0.05$.

For the object collection environment (Fig.5 - a; b), $\xi$-learning outperforms SFQL in both conditions, in tasks with linear and general reward functions. However, the effect is stronger in tasks with general reward functions where SFQL has more problems to correctly approximate the reward function with its linear approach. For the condition, where the agents learn a reward model online (O), the difference between the algorithms in the general reward case is not as strong due the effect of their poor approximated reward models for all agents.

In the racer environment (Fig.5 - c) SFQL has a poor performance below standard Q-learning as it can not appropriately approximate the reward functions with a linear model. In difference CMF $\xi$-learning outperforms QL.

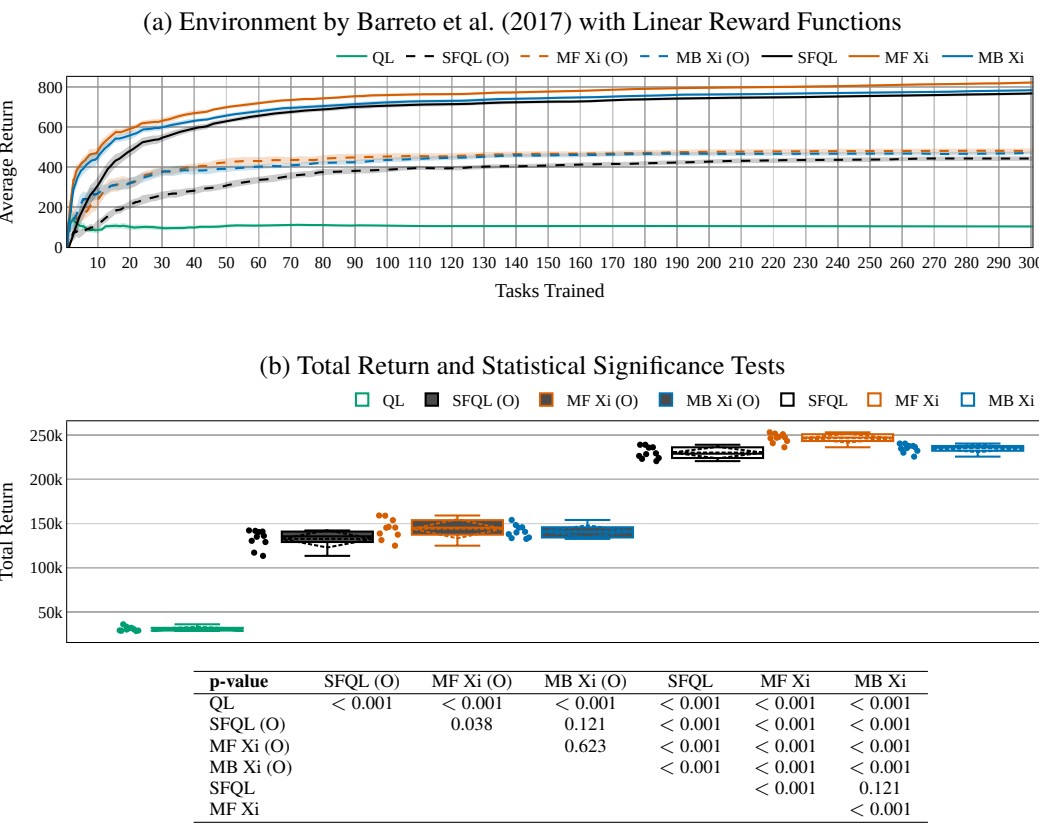

Figure 4: MF $\xi$-learning outperforms SFQL in the object collection environment by Barreto et al. (2017), both in terms of asymptotic performance and learning speed. (a) The average over 10 runs of the average reward per task per algorithm and the standard error of the mean are depicted. (b) Total return over the 300 tasks in each evaluated condition. The table shows the p-values of pairwise Mann–Whitney U tests between the agents.

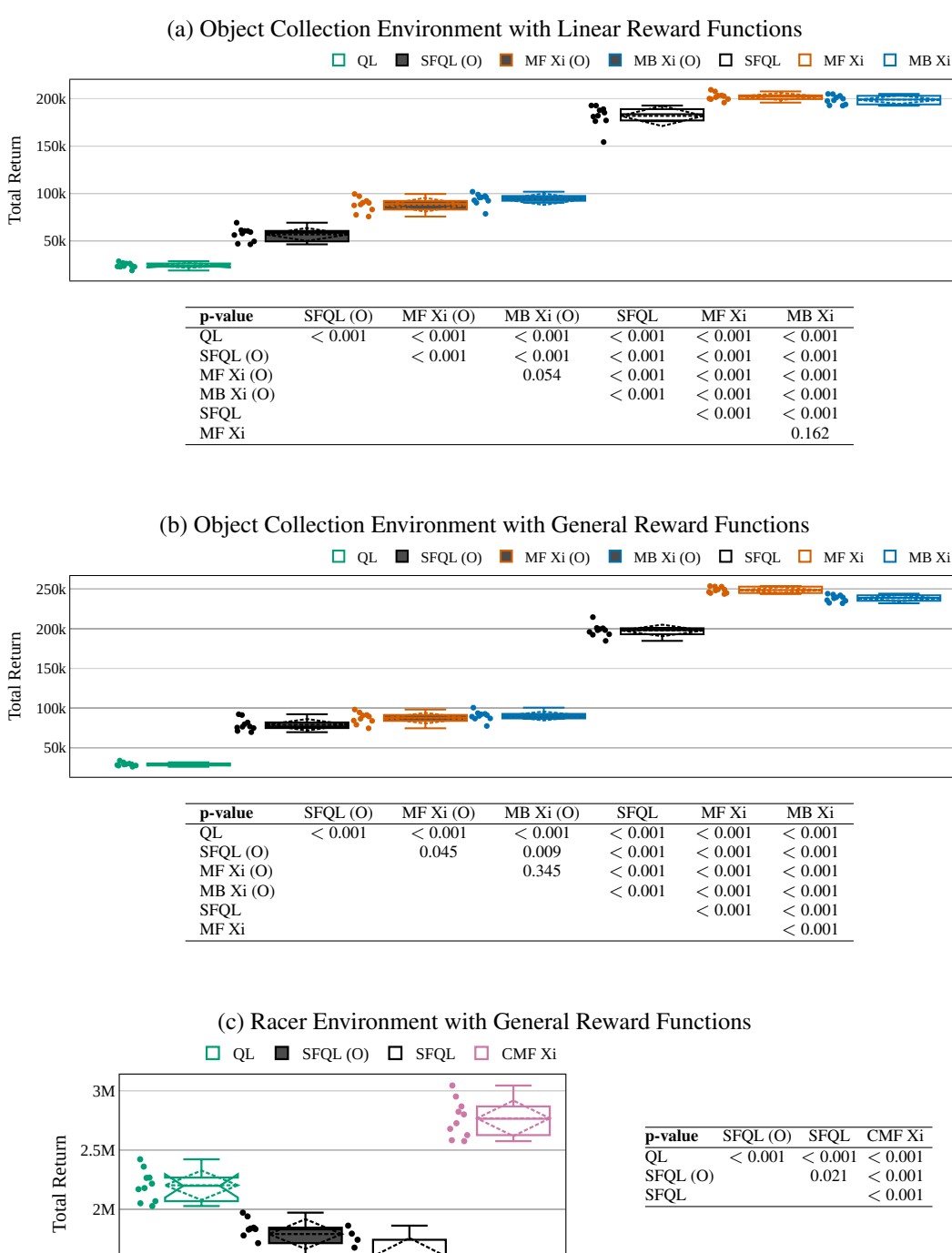

(a) Object Collection Environment with Linear Reward Functions

| p-value | SFQL (O) | MF Xi (O) | MB Xi (O) | SFQL | MF Xi | MB Xi |
|---|---|---|---|---|---|---|
| QL | < 0.001 | < 0.001 | < 0.001 | < 0.001 | < 0.001 | < 0.001 |
| SFQL (O) | | < 0.001 | < 0.001 | < 0.001 | < 0.001 | < 0.001 |
| MF Xi (O) | | | 0.054 | < 0.001 | < 0.001 | < 0.001 |
| MB Xi (O) | | | | < 0.001 | < 0.001 | < 0.001 |
| SFQL | | | | | < 0.001 | < 0.001 |
| MF Xi | | | | | | 0.162 |

(b) Object Collection Environment with General Reward Functions

| p-value | SFQL (O) | MF Xi (O) | MB Xi (O) | SFQL | MF Xi | MB Xi |
|---|---|---|---|---|---|---|
| QL | < 0.001 | < 0.001 | < 0.001 | < 0.001 | < 0.001 | < 0.001 |
| SFQL (O) | | 0.045 | 0.009 | < 0.001 | < 0.001 | < 0.001 |
| MF Xi (O) | | | 0.345 | < 0.001 | < 0.001 | < 0.001 |
| MB Xi (O) | | | | < 0.001 | < 0.001 | < 0.001 |
| SFQL | | | | | < 0.001 | < 0.001 |
| MF Xi | | | | | | < 0.001 |

(c) Racer Environment with General Reward Functions

| p-value | SFQL (O) | SFQL | CMF Xi |
|---|---|---|---|
| QL | < 0.001 | < 0.001 | < 0.001 |
| SFQL (O) | | 0.021 | < 0.001 |
| SFQL | | | < 0.001 |

Figure 5: Total return over all tasks in each evaluated condition. The tables show the p-values of pairwise Mann–Whitney U tests between the agents. See the text for more information.

