# OpenReview forum: "Xi-learning: Successor Feature Transfer Learning for General Reward Functions"
_ICLR.cc/2022/Conference — ICLR 2022 Submitted_

### Official Review · Reviewer_Ccnn · 2021-11-02

**Correctness:** 4
**Technical Novelty And Significance:** 3
**Empirical Novelty And Significance:** 2
**Recommendation:** 5
**Confidence:** 4

**Main Review:**

This paper tackles the important problem of transfer in reinforcement learning. However, the generalization it proposes is oddly positioned. The main premise is described as:

> Nonetheless, this assumption also restricts successful application of
SF&GPI only to problems where such a linear decomposition is possible. This paper investigates the
application of the SF&GPI framework to general reward functions: $r_i = R_i(\phi)$.

This is true in the sense that the linearity of successor features prevents them from representing arbitrary sets of reward functions. (Any individual reward function is representable by setting $\phi_t = r_t$.) One easy example of a reward set that SF could not represent (in continuous spaces) is that of Dirac delta functions over all states, as this would require feature vectors $\phi$ of the same cardinality as the state space. While the proposed $\xi$ variant is indeed more general than the linear SF, these rewards are still functions of $\phi$, so the expressivity of $\xi$ depends on the expressivity of the featurization $\phi.$

One possible response to that criticism is to simply set $\phi_t = \text{concat}(s_t, a_t, s_{t+1})$, which really would fulfill the promise of handling general reward functions. However, this raises a serious practical issue, in that we must now recover the discounted occupancy of a policy in the original state space. This is a pretty difficult combination of dynamic programming and generative modeling (each in their own right hard problems), so is not straightforward to scale (though the paper cites a few attempts). I did not find the discussion of this difficulty too convincing; the main strategy here seemed to rely on a coarse discretization of continuous spaces. I would expect this to work well in the diagnostic environment studied, but mostly because the underlying state is low-dimensional enough that discretizing it and treating it as tabular is a reasonable strategy. (The actual observation is higher-dimensional, but the underlying state looks like position and velocity in $\mathbb{R}^2$.)

I think these two issues put the paper in a strange position:
1. If the feature vectors $\phi$ are unrestricted, then $\xi$ really is more expressive than linear SF, but the method requires solving a long-horizon generative modeling problem to represent the state occupancy and does not propose a viable way of doing this.
2. If the feature vectors $\phi$ are restricted in such a way that $\xi$ is learnable in practice, then $\xi$ likely ends up in the same spot as SF: able to represent any _individual_ reward function, but not any arbitrary set of rewards due to limitations on the featurization itself.

If there were evidence presented that $\xi$-learning is more scalable than I am giving it credit for (_e.g._, compared to SF, in environments for which the linearity assumption is limiting without artificial constraints on the featurization or observation space), then of course I would revise my review.

I want to make sure to list the good too: the paper positions itself well in the SF and GPI literature. While the analysis mostly follows from analogous results in $Q$-learning, it is thorough. There is clean code provided.

**Copy-editing**

*Introduction*:
1. "intput" --> "input"
2. "an cumulative" --> "a cumulative"

*Background*
1. "following following the Bellman equation ": repeated "following"
2. "a MDP" --> "an MDP"
3. "proofed" --> "proved
4. "error in the approximation increase" --> "increases"

*Method*
1. (Prop 1) "quotien space" --> "quotient space"
2. (Corr 2) "supreme norm" --> "supremum norm"?
3. (Prop 2) "well-define" --> "well-defined"

*Experiments*
1. "which object as been already collected" --> "which object has"
2. "number of Gaussian’s": no apostrophe

**Summary Of The Paper:**

This paper proposes a variant of successor features that represents the unnormalized cumulative discounted distribution of state features $\xi(\mathbf{s}, \mathbf{a}) = \sum_t \gamma^t p(\phi_t \mid \mathbf{s}_0 = \mathbf{s}, \mathbf{a}_0 = \mathbf{a})$ instead of the standard expectation $\psi(\mathbf{s}, \mathbf{a}) = \mathbb{E} [ \sum_t \gamma^t \phi_t \mid \mathbf{s}_0 = \mathbf{s}, \mathbf{a}_0 = \mathbf{a}]$. It proposes both a model-free and model-based variant for training $\xi$ and evaluates the effect of non-linear reward functions in two diagnostic environments.

**Summary Of The Review:**

I am in the borderline mode for this paper. Due to the difficulty of the generative modeling problem associated with $\xi$-learning, I am not quite convinced that the generalization presented in this paper would actually lead to something more expressive than SF in the types of setting where the linearity of SF proves limiting. That said, this paper does have merits and adds something new to the SF literature, as mentioned in the main review above.

---

> ### Author Response · Authors · 2021-11-18
> **Author Response**
>
> Thank you for your comments and critiques. These are very insightful and help us to improve our manuscript. We appreciate them a lot.
>
> We are currently working on a rebbutal version of our paper which we will upload at the end of the rebuttal period. Nonetheless, we wanted to respond already to some of your raised points to see if we can answer these for you or if you have further questions.
>
> ---
> > This is true in the sense that the linearity of successor features prevents them from representing arbitrary sets of reward functions. (Any individual reward function is representable by setting $\phi_t = r_t$ .) One easy example of a reward set that SF could not represent (in continuous spaces) is that of Dirac delta functions over all states, as this would require feature vectors of the same cardinality as the state space. While the proposed $\xi$ variant is indeed more general than the linear SF, these rewards are still functions of $\phi$, so the expressivity of $\xi$ depends on the expressivity of the featurization $\phi$.
>
> We agree that $\xi$-learning depends on the featurization $\phi$. Thus it can only optimize reward functions for inputs over the feature space $\Phi$. We do not argue that $\xi$-learning is able to handle arbitry reward functions over the whole state space. Our contribution is that $\xi$-learning can optimize in a more principled way general/arbitrary reward functions over the space $\Phi$ than classical SF Learning. This is also what we showed with our experimental results for general reward functions in the object collection task (Figure 1, c, d) and the racer environment (Figure 3, b).
>
> ---
> > One possible response to that criticism is to simply set $\phi_t = concat(s_t, a_t, s_{t+1})$, which really would fulfill the promise of handling general reward functions. However, this raises a serious practical issue, in that we must now recover the discounted occupancy of a policy in the original state space.
>
> We agree that this formulation would allow us to handle each reward function over the space of transitions ($R(s_t,a_t, s_{t+1})$). This is the original idea of the successor representation from Dayan (1993) from which the SF framework originates. As further pointed out, learning the $\xi$-function over this space is very difficult for high-dimensional problems. But the classical SF framework has the same problem. For this reason, the classical SF framework operates in a space of features that is usually chosen to have a much lower dimension than the dimension of the observations.
>
> Dayan, P. (1993). Improving generalization for temporal difference learning: The successor representation. Neural Computation, 5(4), 613-624.
>
> ---
> > I did not find the discussion of this difficulty too convincing; the main strategy here seemed to rely on a coarse discretization of continuous spaces. I would expect this to work well in the diagnostic environment studied, but mostly because the underlying state is low-dimensional enough that discretizing it and treating it as tabular is a reasonable strategy.
>
> To avoid that there is no misunderstanding: We do not discretize the state space for any of the algorithms (QL, SFQL, or CMF Xi). All algorithms receive as input the continuous observations. For CMF Xi, we use a discretization of the feature space as described in Section 4.2 - Agents (page 8). We do not discretize the feature space for the SFQL agent.
>
> In terms of discretizing the feature space for the $\xi$ agent. Yes, the feature space is low dimensional $\phi \in R^3$. As pointed out in our previous answer the feature space is generally low dimensional in the SF literature.
> What we wanted to show is that $xi$-learning can already outperform classical SFQL in this low-dimensional space.
> Furthermore, given that the discretization takes place for each feature dimension individually, handling larger dimensional spaces should not pose a major problem. For each dimension the xi-values are learned independently.
>
> ---
> > If there were evidence presented that $\xi$-learning is more scalable than I am giving it credit for (e.g., compared to SF, in environments for which the linearity assumption is limiting without artificial constraints on the featurization or observation space), then of course I would revise my review.
>
> In summary, we argue that performing transfer with reward functions that are over the whole transition space $(S, A, S)$ is currently not feasible for many problems as you pointed out. But for the lower dimensional space of features $\Phi$ for which classical SF and $\xi$-learning are intended, $\xi$-learning provides a principled way to handle arbitrary reward functions. Classical SF on the other hand provides linear approximations in this case yielding lower performance as our experimental results show.

---

> > ### Comment · Reviewer_Ccnn · 2021-11-22
> > **Reviewer response**
> >
> > Thanks for the follow-up; I have read this response and the general response above.
> >
> > I think the main issue of this particular generalization not being sufficiently motivated still remains. While it is possible to design features adversarially such that linear SF cannot represent the rewards, it would be much more valuable and convincing if there were a reason to use this generalization even when features could be learned. I agree with your point in the general response that learning features does add complexity, but arguably learning $\phi$ is much more straightforward than learning $\psi$ or $\xi$ (due to RL being used for the latter), so it does not seem like the main bottleneck.
> >
> > One minor question about the scaling of the discretization:
> > > Furthermore, given that the discretization takes place for each feature dimension individually, handling larger dimensional spaces should not pose a major problem. For each dimension the xi-values are learned independently.
> >
> > Does that mean that the $\xi$ cannot represent the joint distribution over features (and only the product of per-dimension marginals)?

---

> > > ### Author Response · Authors · 2021-11-22
> > > **Authors Response**
> > >
> > > Thank you for your continuing interest.
> > >
> > > ---
> > > >  While it is possible to design features adversarially such that linear SF cannot represent the rewards, it would be much more valuable and convincing if there were a reason to use this generalization even when features could be learned.
> > >
> > > As we pointed out in our second point from our general response, our replication (Sec. E.1) of the object collection task from Barreto et al. (2017) shows the performance of learned features is below the performance of given features.
> > > MF Xi reaches a final average reward per task of 850 with given features and reward functions.
> > > SFQL with learned features (8 dimensions) reaches a significant lower final performance of 575 (Fig. 2 in (Barreto et al., 2017)).
> > >
> > > Thus, although features can be learned in this task, they seem not to be as beneficial as $\xi$-Learning with given features and reward functions.
> > >
> > > ---
> > > > but arguably learning $\phi$ is much more straightforward than learning $\psi$ or $\xi$ (due to RL being used for the latter), so it does not seem like the main bottleneck
> > >
> > > We are a bit confused here, as even in classical SF, if we learn the features $\phi$ we also have to learn $\psi$. We have to learn $\psi$  (classical SF) or $\xi$ ($\xi$-learning) regardless of the question if we learn the features or not.
> > > And our results show that learning $\xi$ has a better performance (for linear and general reward functions).
> > >
> > > ---
> > > > Does that mean that the $\xi$ cannot represent the joint distribution over features (and only the product of per-dimension marginals)?
> > >
> > > The general procedure of $\xi$-learning can handle the joint distribution in terms of our theoretical results, e.g. its convergence proof.
> > > Our particular implementation of CMF Xi for the continuous feaure environment is using the fact that the reward functions in this case are  a sum of independent reward components: $r(\phi) = \sum_k r_k (\phi_k)$. We discuss this situation in our discussion section about "Continuous Feature Spaces" on page 8.
> > >
> > > ---
> > > Thanks again for pointing out the several grammatical errors. We fixed them now in our new rebuttal pdf version of the paper.

---

### Official Review · Reviewer_wZjF · 2021-11-02

**Correctness:** 3
**Technical Novelty And Significance:** 2
**Empirical Novelty And Significance:** 2
**Recommendation:** 5
**Confidence:** 4

**Main Review:**

On the writing:
The paper is nicely written and well structured, making it a pleasant read. It also appears to be technically sounds.

On the proposed method:
My main concern is that authors didn't acknowledge previous efforts by Barreto et al. 2018 to address the problem of general reward functions. The authors state that the upper-bound in (7) is only interesting if the reward function can be decomposed linearly with respect to state-action features. Barreto et al. 2018 proposed a principled way of learning features (or even the reward function) such that the approximation error on the reward remains low, and the upper-bound on the optimal Q-value error remains low as well.

Without a comparison with Barreto et al. 2018 method, it is difficult to understand the necessity of introducing a new quantity to learn (the expected cumulative discounted probability of SF). While I like the formalism and the theoretical guarantees, to recommend the paper to be accepted, I would need more convincing on the necessity of the xi-function, specially given how difficult it will be to learn in large and stochastic environments.

On the experiment:
- Authors should specify how the weights \tilde{w_i} were learned for SFQL in the general case.
- Results including the learning of the reward function - or weights w - would be very interesting to appear in the main text, as in practice we rarely have access to the reward function. It would help build a case for practical use of the proposed method.
- It seems to me that the continuous environment is rather a discretized environment?
- I did not understand the proposed justification for xi-learning improving on SFQL (even slightly) in environments with linear reward function. What did the authors mean by "xi-learning reduces the complexity for the function approximation of the xi-function compared to the phi-function in SFQL"?

Minor comment: I found the second sentence of Theorem 1 difficult to understand.


**Summary Of The Paper:**

This paper addresses a limitation of the successor features (SF) which were introduced as a mechanism for transfer learning in reinforcement learning when the reward function changes across tasks. The authors claim that the original SF framework will only provide a good approximation of the true Q-function if the reward function in a task can be represented using a linear decomposition, and propose to address this linear decomposition of the reward function requirement by introducing a novel SF mechanism, Xi-learning, based on learning a cumulative discounted probability of successor features. The paper includes theoretical proofs of the convergence of xi-learning as well as transfer learning guarantees under generalized policy improvement (GPI). The authors compare the performance of their methods against standard Q-learning and SFQL in two different domains.

**Summary Of The Review:**

Despite enjoying the reading of the paper, the main reason for my recommendation is that due to the lack of discussion / comparison with another method that adresses the non-linearity of the reward function to learn accurate SF, I am not convinced by the necessity of the xi-function, which is the novelty of the paper.

---

> ### Author Response · Authors · 2021-11-18
> **Author response**
>
> Thank you for your comments and critiques. These are very insightful and help us to improve our manuscript. We appreciate them a lot.
>
> We are currently working on a rebbutal version of our paper which we will upload at the end of the rebuttal period. Nonetheless, we wanted to respond already to some of your raised points to see if we can answer these for you or if you have further questions.
>
> ---
> > Barreto et al. 2018 proposed a principled way of learning features (or even the reward function) such that the approximation error on the reward remains low, and the upper-bound on the optimal Q-value error remains low as well.
> Without a comparison with Barreto et al. 2018 method, it is difficult to understand the necessity of introducing a new quantity to learn ...
>
> We agree that learned features are in principle able to handle arbitrary reward functions, but these seem to not be able to perform as well as given features. Please note the second point of our common reply about learning of features.
>
> ---
> > Authors should specify how the weights \tilde{w_i} were learned for SFQL in the general case.
>
> Thank you for noting that we missed giving a more precise description of it in the appendix. We will fix it for the rebuttal version which we will upload at the end of the rebuttal period.
>
>
> ---
> > Results including the learning of the reward function - or weights w - would be very interesting to appear in the main text, as in practice we rarely have access to the reward function. It would help build a case for practical use of the proposed method.
>
> This decision was mainly done for the reason of restricted space, as it would then require to describe all the results also in the main text. As we are currently mainly considering non-learned features and reward functions we believe that the results for these are more important. Please see also our common reply on the learning of features and especially the first point about problems where we have access to rewards and features.
>
> ---
> > It seems to me that the continuous environment is rather a discretized environment?
>
> To make sure there is no misunderstanding: We do not discretize the state space for any of the algorithms (QL, SFQL, or CMF Xi). All algorithms receive as input the continuous observations. For CMF Xi, we use a discretization of the feature space as described in Section 4.2 - Agents (page 8). We do not discretize the feature space for the SFQL agent.
>
>
> ---
> > I did not understand the proposed justification for xi-learning improving on SFQL (even slightly) in environments with linear reward function. What did the authors mean by "xi-learning reduces the complexity for the function approximation of the xi-function compared to the phi-function in SFQL"?
>
> For the linear case, classical SF and Xi-learning are equivalent in their ability to compute the correct Q-values. Therefore, the only difference that we see between both algorithms is in their way to approximate their respective functions, and as Xi-learning yields a better performance its function approximation seems to be easier, i.e. less complex.

---

> > ### Author Response · Authors · 2021-11-23
> > **Author Response**
> >
> > Further comments in regard to our final rebuttal version of the paper.
> >
> > > Authors should specify how the weights \tilde{w_i} were learned for SFQL in the general case.
> >
> > The procedure is explained in Section C.3 (page 20). We added now references to this section from the sections were we describe the SFQL algorithm.

---

### Official Review · Reviewer_wA2K · 2021-11-03

**Correctness:** 3
**Technical Novelty And Significance:** 3
**Empirical Novelty And Significance:** 2
**Recommendation:** 5
**Confidence:** 3

**Main Review:**

The authors propose an interesting extension to linear SF framework. The derivations look correct to me and are overall well written.

Just a few suggestions on the presentation of the method.
- It would be helpful if you reduce the new formulation to the special case linear SF around equation (13) and illustrate the key difference.
- discuss the difficulty of simultaneously learning features and estimating the density of features.

Kudos to the authors conducted u-test for the significance of the results. Appreciated the effort in making more rigorous results.

The primary weakness of this work is the experimental section.
There seems to a number of different issues all conflated into one set of experiments.

Firstly, in both collection and racer environments, the features are manually defined instead of learning from observations. This brings up a major concern that it hurts the flexibility and expressiveness of linear SF.
With learned features, even if we assume reward is a linear function of features, the resulting value function is not hindered and in fact the expressiveness of value function is not limited by the linearity, as features can be an expressive and arbitrarily non-linear function of observations.

Secondly, it seems in all experiments the evaluation tasks set is the same as the training tasks set. This makes make it hard to interpret the experiments or understand how effective is the transfer. I encourage the authors to divide the tasks into training set and hold out test set, and show the average return on the hold out set.

Lastly, I believe the authors should compare again random feature baselines, where a neural network is randomly initialized and fixed during training, and then running linear SF and xi-learning on top it. Similar to Hansen et al. This would provide insights on how the SF works in these domains.

These issues are probably fixable by more careful experimentation and clarity on the exact setup.

A general weakness is that the tasks are very simple, to the point that it is very unclear why the linear SF formulation does worse than xi-learning. I encourage the authors to try to evaluate the method at least on the Mujoco tasks from Barreto et al 2018.

A question about the reward definition. The rewards for collection environment is defined as “The general reward functions are sampled by assigning a different reward to each possible combination of object properties ..., such that picking up an orange box might result in a reward of ...”, I don’t fully understand the rationality behind the particular choice, could the authors elaborate more?

**Summary Of The Paper:**

The authors extend the successor features (SF) linear formulation to general non-linear formulation. The new method allows the reward to be arbitrary composition of features.

The propose method is based on learning cumulative discounted probability of SF rather than cumulative discounted sum of SF as in linear SF framework.
The authors provide theoretical proofs about convergence and propose two practical methods. Experiments are conducted in two toy environments.


**Summary Of The Review:**

In summary, the paper is an interesting effort in extending the linear SF framework.

But why the linear assumption in SF framework is bad is not well supported, especially when the features can be an expressive and arbitrarily non-linear function of observations.

The experiments failed to show very convincing results due to simple environments and problematic experimentation setup.

The authors are encouraged to address these concerns. It would be a nice contribution if the concerns are resolved.

---

> ### Author Response · Authors · 2021-11-18
> **Author Response**
>
> Thank you for your comments and critiques. These are very insightful and help us to improve our manuscript. We appreciate them a lot.
>
> We are currently working on a rebbutal version of our paper which we will upload at the end of the rebuttal period. Nonetheless, we wanted to respond already to some of your raised points to see if we can answer these for you or if you have further questions.
>
>
> ---
> >It would be helpful if you reduce the new formulation to the special case linear SF around
> equation (13) and illustrate the key difference.
>
> Yes, it is possible to show that in the case of linear reward functions the Xi-learning framework can be reduced to the classical SF formulation. This shows that Xi-learning can represent and solve environments where a linear assumption holds similar to SFQL. And it shows that Xi-learning extends the SF framework to general reward functions. We will include this discussion on the rebuttal version of our paper which we will upload at the end of the rebuttal period.
>
>
> ---
> > discuss the difficulty of simultaneously learning features and estimating the density of features.
>
> We are currently not learning features simultaneously while learning the Xi-function (density of features). This is also not done in the original work by Barreto (2017) or Barreto (2019). Instead, the features are learned offline before the learning of the psi functions.
> Please note also our common reply on the topic learning features that we added. For the moment we feel that the learning of features is outside the scope of the paper.
>
> ---
> > Firstly, in both collection and racer environments, the features are manually defined instead of learning from observations. This brings up a major concern that it hurts the flexibility and expressiveness of linear SF. With learned features, even if we assume reward is a linear function of features, the resulting value function is not hindered and in fact the expressiveness of value function is not limited by the linearity, as features can be an expressive and arbitrarily non-linear function of observations.
>
> Please note our common reply on the topic of learning of features.
>
> ---
> > Secondly, it seems in all experiments the evaluation tasks set is the same as the training tasks set. This makes make it hard to interpret the experiments or understand how effective is the transfer. I encourage the authors to divide the tasks into training set and hold out test set, and show the average return on the hold out set.
>
> We follow the experimental procedure that was introduced by Barreto (2017) and used in other SF literature to have the best comparison to the classical SF framework. The experiments are conducted in a continual learning setting where we introduce new tasks to the agent as learning progresses. The learning curves with the average return per task, for example in Figure 1 (b, d), show how the methods are able to use their knowledge from previous tasks to new ones compared to the other methods.
>
>
> ---
> > Lastly, I believe the authors should compare again random feature baselines, where a neural network is randomly initialized and fixed during training, and then running linear SF and xi-learning on top it. Similar to Hansen et al. This would provide insights on how the SF works in
> these domains.
>
> This is indeed an interesting question when it comes to the question of learned features. Then it will be interesting to test how learned features will perform against random features. For the moment we do not have investigated learned features. Please see our common reply on the topic of learning features.
>
> ---
> > A general weakness is that the tasks are very simple, to the point that it is very unclear why the linear SF formulation does worse than xi-learning. I encourage the authors to try to evaluate the method at least on the Mujoco tasks from Barreto et al 2018.
>
> We agree that this is an interesting experiment, but unfortunately due to limited time constraints, we are not able to run it for the rebuttal.
> Nonetheless, we still believe that the three environments that we use (the modified object collection task, the original object collection task, and the racer environment) provide enough evidence for the interest in xi-learning, as these cover stochastic environments with continuous observations (with around 120 dimensions) and discrete and continuous features.

---

> > ### Author Response · Authors · 2021-11-23
> > **Author Response**
> >
> > We forgot to answer the following question in our last reply.
> >
> > >A question about the reward definition. The rewards for collection environment is defined as “The general reward functions are sampled by assigning a different reward to each possible combination of object properties ..., such that picking up an orange box might result in a reward of ...”, I don’t fully understand the rationality behind the particular choice, could the authors elaborate more?
> >
> > We use this procedure to generate general reward functions instead of linear reward functions. The objects in our adapted task have 2 properties each with 2 possible values: color (orange or blue) and form (triangle or box). Similar to the original object collection task from Barreto et al., we are using a binary vector to represent these features with $\phi = [$is_orange, is_blue, is_triangle, is_box$]^\top \in [0,1]^4$  (please note that we are ignoring here the last feature for reaching the goal area). We could now define general reward functions by using non-linear functions that incorporate for example multiplications, trigonometry functions, or other non-linear operations over the feature vector, for example $r = \prod_{i=1}^4 \phi_i$.
> >
> > Instead we used our method which randomly samples for each task a specific reward for each combination of features: r(blue box), r(blue triangle), r(orange box), and r(orange triangle). This allows us to sample in the whole function space over all possible reward functions (general and linear) that create rewards that are bounded in [-1,1]. As we stated reward functions sampled according to this procedure can in most cases not be linearily composed into $r(\phi) = \phi^\top \cdot \mathbf{w}$.

---

### Official Review · Reviewer_UrNN · 2021-11-08

**Correctness:** 3
**Technical Novelty And Significance:** 3
**Empirical Novelty And Significance:** 2
**Recommendation:** 5
**Confidence:** 3

**Main Review:**

The motivation of this paper is not described and supported well. Why the assumption of linear composition is bad, or not applicable to what domains? From the results, previous SF methods perform not that badly on the problem with a general reward function.
It is more suitable to say two environments with discrete state space and continuous state space.


For experiments,
Why not compare with [1], and some SOTA meta RL algorithms?

For the second environment, why discretize the state space, since deep SF can be applied to continuous domains?

The reason that why MF \xi-learning outperforms MB \xi-learning is not discussed. Since MF \xi-learning is superior, what is the need to propose MB \xi-learning?

Results of MB \xi-learning in the second environment are not provided.

Some descriptions need to be improved. MB \xi-learning refers to Model-based \xi-learning, not Modle \xi-learning.


[1] Fast reinforcement learning with generalized policy updates.


**Summary Of The Paper:**

This paper proposes a new successor feature learning algorithm, called \xi-learning. Previous SF assumes reward function is a linear composition of SF and reward weights, this paper extends previous SF to a setting with a general reward function over SF. Based on this, \xi-learning learns to estimate the cumulative discounted probability of SF, and provides two update operators, model-free \xi-learning and model-based \xi-learning. The convergence of \xi-learning is proved, and empirical results on environments with discrete and continuous state spaces show it outperforms previous SF methods.

**Summary Of The Review:**

I think this paper contributes to the SF research areas, but the motivation is not well discussed. Also, there are some details, analysis of results should be well discussed. Based on this, I give a borderline reject currently.

---

> ### Author Response · Authors · 2021-11-18
> **Author response**
>
> Thank you for your comments and critiques. These are very insightful and help us to improve our manuscript. We appreciate them a lot.
>
> We are currently working on a rebbutal version of our paper which we will upload at the end of the rebuttal period. Nonetheless, we wanted to respond already to some of your raised points to see if we can answer these for you or if you have further questions.
>
>
> ---
> >The motivation of this paper is not described and supported well. Why the assumption of linear composition is bad, or not applicable to what domains? From the results, previous SF methods perform not that badly on the problem with a general reward function. It is more suitable to say two environments with discrete state space and continuous state space.
>
> Classical SF with its linear composition works of course well if this assumption holds.
> The object collection task by Barreto et al. (2017) is a clear example, as the weight vector can precisely define the reward for each object type that is collected.
>
> Even in environments where a linear composition does not exist, the classical SF framework can still approximate the reward function and therefore perform well for some reward functions. This is shown in Figure 2 (c), where reward functions that are nearly linear composable lead to a similar performance of SFQL and MF Xi. But as the reward functions violate this assumption more and more, SFQL can only reach 50% of the performance of MF Xi.
>
>
>
>
> ---
> > For experiments, Why not compare with [1], and some SOTA meta RL algorithms?
>
> Our goal with this seminal paper is to introduce Xi-learning as a potential alternative to classical SF&GPI and to open with it a new line of research. We therefore decided to compare it directly to its closest algorithm in the classical SF&GPI framework which is the original algorithm by Barreto et al. (2017).
>
> ---
> >For the second environment, why discretize the state space, since deep SF can be applied to continuous domains?
> Please note, we do not discretize the state space for any of the algorithms (QL, SFQL, or CMF Xi). All algorithms receive as input the continuous observations. For CMF Xi, we use a discretization of the feature space as described in Section 4.2 - Agents (page 8). We do not discretize the feature space for the SFQL agent.
>
> We agree that the paragraph that describes this is very short and prone to misunderstandings. We try to improve this in our rebuttal version which we upload at the end of the rebuttal period. Thank you.

---

> > ### Author Response · Authors · 2021-11-23
> > **Author Response**
> >
> > Further comments in regard to our final rebuttal version of the paper.
> >
> > ---
> > > The reason that why MF \xi-learning outperforms MB \xi-learning is not discussed. Since MF \xi-learning is superior, what is the need to propose MB \xi-learning?
> >
> > We added MB xi-learning as an ablation study. Although it does not outperform MF Xi in most cases (see Figure 5 - a, page 30 for an exception where MB Xi (O) it outperforms MF Xi (O) in the object collection task with linear reward functions) we still thought it useful for readers to its performance, in case someone wanted to implement and test such an extension to Xi-learning.
> >
> > We now moved the description of MB Xi to the appendix to save space for a further discussion of learned features.
> >
> > ---
> > > Results of MB \xi-learning in the second environment are not provided.
> >
> > We did not have the time yet to run these experiments, but we intend to add the result for a camera-ready version.
> >
> > ---
> > > Some descriptions need to be improved. MB \xi-learning refers to Model-based \xi-learning, not Modle \xi-learning.
> >
> > We refer for the procedure now as One-Step Model-based (MB) $\xi$-Learning.

---

### Author Response · Authors · 2021-11-18
**Learned features / reward weights vs predefined features / reward functions**

The reviewers pointed out that classical SF&GPI can also handle arbitrary reward functions if the features are learned and therefore asked for a comparison of Xi-Learning to SFQL with learned features. We thank the reviewers for pointing out that we missed to discuss this point. We will include a discussion in the rebuttal version of the paper which we will upload at the end of the rebuttal period. We are providing here already our thoughts to start a possible discussion before the end of the rebuttal period.

We agree with the reviewers that learned features together with learned reward weights are in principle able to represent and handle non-linear reward functions. This is only possible if the learned features describe the non-linear effects in the reward functions.

Nonetheless, we still believe that focusing on predefined features and reward functions provides sufficient evidence for the interest in Xi-Learning. We base this on three main arguments: 1) Features and reward functions are known for many problems with non-linear reward functions, 2) learning of features seems not to improve performance, and 3) learning of features adds further challenges:

1) For many RL problems the features and the reward functions of the tasks are known. This is for example the case for most robotic tasks where the reward functions are defined and optimized by humans (Akalin & Loutfi, 2021). Thus using pregiven features and reward functions is natural for many applications.

2) The performance of learned features / reward weights seems below the performance of pregiven features / reward functions as shown by the original experiment by Barreto (2017) (Figure 2) and our replication (Appendix D.1) where pregiven features / reward functions perform better (with a factor of ~1.8) than learned features. The results by Barreto et al. (2017) show that learned features with a dimension of h=4 result in the same performance as SFQL with given features (4 dimensions). Learned features with a dimension of h=8 improve the performance, reaching a final average reward per task performance of ~575 after 250 tasks (Fig 2, Barreto et al. [2017]).
In comparison, using the given features and the given reward functions yields ~750 for SFQL and ~850 for MF Xi learning. Using predefined features and reward functions together with Xi-learning seems therefore even better in linear cases than classical SFQL with learned features.

3) Learning features adds extra complexity to the learning process. In principle, two options are possible to learn them: a) prelearning, or b) continuously learning.
a) Prelearning requires collecting data from several tasks to learn the features before being able to use them with the SF&GPI procedure. This reduces its performance in the initial tasks and adds extra learning effort. Furthermore, the initial tasks, i.e. reward functions, from which we sampled the data to learn the feature might not include certain non-linearities which might be introduced by much later tasks. Thus, the SFQL agent could not represent these nonlinear reward functions well.
b) Continuously learning them alongside the SF&GPI procedure introduces complexity in how to manage both learning processes. First, the feature learning should be most probably slower than the learning of the psi/xi-functions as these depend on the features. Second, psi/xi-functions from previously learned tasks can not be directly applied to the GPI procedure. This is because the currently learned reward weights depend on the current feature representation which are incompatible for the feature representation on which the old psi/xi-functions are trained. An extra mechanism is needed to update the psi/xi-functions according to the continuously changing features or to compensate for the effect of changing features.
In summary, the learning of features is a complex problem by itself that is both true for classical SF and Xi-Learning.

In conclusion, our goal with this seminal paper is to introduce Xi-learning as a potential alternative to classical SF&GPI and to open with it a new line of research. We believe therefore that the learning of features is currently out of scope of the paper.

---
Akalin, N., & Loutfi, A. (2021). Reinforcement learning approaches in social robotics. Sensors, 21(4), 1292.

A. Barreto, W. Dabney, R. Munos, J. J. Hunt, T. Schaul, H. P. van Hasselt, and D. Silver. (2017) Successor features for transfer in reinforcement learning. In Advances in neural information processing systems, pages 4055–4065

---

### Author Response · Authors · 2021-11-23
**New Rebuttal Version - 22.11.2021**

We uploaded a new version of the paper with the following main changes:

 * New results for the racer environment (Fig. 2 - b, page 7) with an adapted version of CMF Xi. The adapted version is able to significantly outperform QL and SFQL. The new CMF Xi version uses a deeper architecture (2 hidden layers with 20 neurons) and an improved update operation (see Section D.2.3, page 25 for more information). The QL and SFQL algorithms were also adapted to be comparable.

 * An added discussion about the learning of features (under Section 5, page 8) reiterating our points from our previous post here.

 * We moved the description of the One-Step SF Model-based (MB) Xi-Learning procedure to the appendix (Section B, page 15) to save space for the discussion of feature learning.

 * We added an appendix (Section A.6, page 15) that shows that classical SF is a special case of Xi-learning in the case of tasks with linear reward functions.

---

### Decision · Program_Chairs · 2022-01-20

**Decision:**

Reject

**Comment:**

The reviewers uniformly suggested rejecting the current paper.

I concur and remain especially somewhat unconvinced by the authors comments on learning features.  In particular, any argument based simply on (current) performance seems rather weak.  There are methodological reasons one might want to keep features fixed, and there are a small subset of problems with well-defined known useful features.  But in the long term surely we should want to be able to learn the features, and efficiently and elegantly handle the case where they are learnt continually.

I want to thank the authors for engaging.  This work has the potential to be improved and I would encourage the authors to carefully consider and incorporate the provided feedback by the reviewers into their work.